# Constitutive smooth muscle tumour necrosis factor regulates microvascular myogenic responsiveness and systemic blood pressure

Jeffrey T. Kroetsch[1,2], Andrew S. Levy[1,2,3], Hangjun Zhang[1], Roozbeh Aschar-Sobbi[4], Darcy Lidington[1,2], Stefan Offermanns[5,6], Sergei A. Nedospasov[7,8], Peter H. Backx[4,9,10], Scott P. Heximer[1,9] & Steffen-Sebastian Bolz[1,2,3,9]

Tumour necrosis factor (TNF) is a ubiquitously expressed cytokine with functions beyond the immune system. In several diseases, the induction of TNF expression in resistance artery smooth muscle cells enhances microvascular myogenic vasoconstriction and perturbs blood flow. This pathological role prompted our hypothesis that constitutively expressed TNF regulates myogenic signalling and systemic haemodynamics under non-pathological settings. Here we show that acutely deleting the TNF gene in smooth muscle cells or pharmacologically scavenging TNF with etanercept (ETN) reduces blood pressure and resistance artery myogenic responsiveness; the latter effect is conserved across five species, including humans. Changes in transmural pressure are transduced into intracellular signals by membrane-bound TNF (mTNF) that connect to a canonical myogenic signalling pathway. Our data positions mTNF 'reverse signalling' as an integral element of a microvascular mechanosensor; pathologic or therapeutic perturbations of TNF signalling, therefore, necessarily affect microvascular tone and systemic haemodynamics.

[1] Department of Physiology, Faculty of Medicine, University of Toronto, 1 King's College Circle, Medical Sciences Building, Toronto, Ontario, Canada M5S 1A8. [2] Toronto Centre for Microvascular Medicine at TBEP, University of Toronto, 661 University Avenue, 14th floor, Toronto, Ontario, Canada M5G 1M1. [3] Keenan Research Centre at St. Michael's Hospital, 30 Bond Street, Toronto, Ontario, Canada M5B 1W8. [4] Division of Cardiology, University Health Network, R. Fraser Elliott Building, 1st Floor, 190 Elizabeth Street, Toronto, Ontario, Canada M5G 2C4. [5] Max-Planck-Institute for Heart and Lung Research, Ludwigstrasse 43, 61231 Bad Nauheim, Germany. [6] Centre for Molecular Medicine, University of Frankfurt, Theodor-Stern-Kai 7, Frankfurt am Main 60590, Germany. [7] Engelhardt Institute of Molecular Biology and Lemonosov Moscow State University, 32 Vavilov Street, Moscow 119991, Russia. [8] German Rheumatism Research Center, a Leibniz Institute, Chariteplatz 1, Berlin 10117, Germany. [9] Heart & Stroke/Richard Lewar Centre of Excellence for Cardiovascular Research, University of Toronto, C. David Naylor Building, 6 Queens Park Cresc. West, Toronto, Ontario, Canada M5S 3H2. [10] Department of Biology, York University, Farquharson Building, 110 Campus Walk, Toronto, Ontario, Canada M3J 2S5. Correspondence and requests for materials should be addressed to S.-S.B. (email: sts.bolz@utoronto.ca).

Tumour necrosis factor (TNF) is primarily characterized as an immunomodulatory cytokine[1]. Its ubiquitous expression[2], however, implies important functions beyond the immune system. Our previous work in cerebral and mesenteric arteries demonstrates that smooth muscle cell-derived TNF is central to the pathological augmentation of myogenic reactivity[3–5]; it enables systemic diseases with the ability to alter microvascular autoregulation and hence, local tissue perfusion. Accordingly, sequestering TNF with etanercept (ETN) normalizes pathologically enhanced myogenic responsiveness in cerebral arteries[3–5]. In mesenteric arteries, however, the same intervention abolishes myogenic reactivity[5], fuelling the hypothesis that TNF may serve as a constitutive regulator of myogenic signalling in certain microvascular beds. If this observation extends to skeletal muscle resistance arteries, which prominently regulate total peripheral resistance (TPR)[6], TNF would emerge as a physiological regulator of mean arterial pressure (MAP). The immediate clinical ramification would be that anti-TNF therapy must be more cautiously applied and monitored, since it could possess an unappreciated capability to modulate TPR and hence, haemodynamic parameters. Indeed, anti-TNF therapeutics have cardiovascular side effects that are incompletely understood[7].

The biology of TNF is complex[8–10]. TNF is expressed on the cell surface as membrane-bound form (26 kDa) that can be cleaved by extracellular matrix metalloproteinases (TNF converting enzyme; TACE) to yield a soluble form (17 kDa) (refs 11,12). The membrane-bound (mTNF) and soluble (sTNF) forms of TNF exist as homotrimers[13] that are both biologically active and partially overlap in function[10,14]. sTNF activates well-described signalling pathways via two receptors (TNF receptor 1; TNFR1/TNF receptor 2; TNFR2)[15–17]. For mTNF, both forward (via receptors) and reverse (outside-in signalling through the TNF protein) signalling pathways have been described[10]. Of the options available, an mTNF signal makes intuitive sense: since myogenic signalling is perpetually active, proteolytic shedding (that cleaves mTNF to release sTNF) would rapidly deplete cellular TNF and is, therefore, unsustainable.

In this report, we show that acutely deleting the TNF gene in smooth muscle cells or scavenging TNF with the clinical therapeutic ETN reduces resistance artery myogenic responsiveness and hence, systemic blood pressure. The inhibitory effect on myogenic responsiveness is conserved across five distinct species, including humans. Changes in transmural pressure are transduced into intracellular signals by membrane-bound TNF (mTNF), thereby positioning mTNF as a mechanosensor; this 'reverse signal' connects to the established intracellular myogenic signalling elements extracellular signal regulated kinases 1 and 2 (ERK1/2) and sphingosine kinase 1 (Sphk1). Our data, therefore, suggest that several side effects of clinical ETN therapy result from interfering with TNF's pivotal mechanosensor function and hence, microvascular tone and systemic haemodynamic. Thus, perturbing TNF signalling can potentially elicit deleterious effects in the cardiovascular system.

## Results

### Etanercept abrogates the myogenic response in heart failure.
Consistent with our hypothesis that TNF constitutively drives myogenic signalling in skeletal muscle resistance arteries, ETN abolishes myogenic responsiveness in an experimental mouse model of heart failure (HF; Fig. 1a,b), rather than simply normalizing it. Unfortunately, germline TNF knockout mice ($Tnf^{-/-}$ mice) do not provide an appropriate genetic correlate of pharmacological TNF inhibition: the myogenic mechanisms adapt to the inherited loss of TNF activity through the recruitment of alternative cellular signalling pathways[18]. Thus $Tnf^{-/-}$ and wild-type mice have quantitatively similar myogenic responses (Fig. 1c,d)[4,5,19] and systemic haemodynamics[20] under non-pathological settings.

### Acute TNF deletion reduces myogenic tone and blood pressure.
To bypass the compensatory adaptations that occur following germline $Tnf$ gene deletion, we employed a model of acute smooth muscle cell $Tnf$ gene removal (that is, 3 days of tamoxifen treatment in SMMHC-$Cre$ER$^{T2}$ mice[21] crossed with $Tnf^{flox/flox}$ mice[22], generating Sm-TNF-KO[4,5]). Although MAP is normal in anaesthetized Sm-TNF-KO mice (Fig. 2a), further analysis reveals a 20% reduction TPR (Fig. 2b) and a concomitant 16% increase in cardiac output (CO, Fig. 2c). The latter is driven by a combination of elevated heart rate and cardiac contractility (Supplementary Tables 1 and 2) and is a typical, sympathetically mediated compensatory response to reduced TPR. We therefore capitalized on well-characterized diurnal fluctuations (that is, activity vs rest phase) in sympathetic activity[23] and cardiac performance[24,25] to further segregate our MAP analysis. We predicted that the capacity to compensate for reduced TPR should be minimized when sympathetic activity is lowest (that is, during the light/rest phase[26,27]). Indeed, telemetric measurements demonstrate that Sm-TNF-KO have reduced MAP in the light phase (20% reduction at day 4 of tamoxifen treatment, Fig. 2d,e), without overt heart rate compensation (Supplementary Fig. 1) or differences in ambulatory activity (Fig. 2f). In the dark/active phase, the restoration of normal MAP (Fig. 2g–i) is driven by a heart rate-dependent increase in CO (Supplementary Fig. 1b).

The progressive reduction of MAP in tamoxifen-treated $Tnf^{flox/flox}$ mice (the decline in mean, systolic, and diastolic blood pressures in Sm-TNF-KO have statistically significant non-zero slopes; $P < 0.05$ in a F-test; control slopes are statistically zero in a F-test, Fig. 2j–l) strongly associates with the progressive attenuation of myogenic responsiveness in isolated skeletal muscle resistance arteries (at days 2 and 3, Fig. 2m,n). This positions myogenic responsiveness, a significant contributor to TPR[28], as the root cause for the haemodynamic changes. The effect of silencing TNF signalling is specific to myogenic responsiveness, since receptor-dependent vasoconstriction is not altered (Supplementary Fig. 2). We therefore conclude that TNF is a constitutive regulator of skeletal muscle resistance artery myogenic vasoconstriction and hence, TPR and MAP.

### Etanercept reduces blood pressure and myogenic signalling.
Our Sm-TNF-KO data predict that pharmacological TNF inhibition $in$ $vivo$ should alter systemic haemodynamic parameters and lower MAP levels in the rest/light phase. Indeed, ETN injection (20 mg kg$^{-1}$, intraperitoneal) induces a rapid $\sim$50 mm Hg drop in MAP in the light/rest phase, without reducing heart rate (Fig. 3a,b); this MAP reduction also occurs in mice with HF (Fig. 3c,d). To confirm that ETN induces haemodynamic changes by specifically interacting with TNF, we disrupted ETN's structure (heat denaturation) and thereby abolished its haemodynamic effect (Fig. 3e,f). Injections of equal volumes of saline do not affect either MAP or heart rate (Fig. 3g–l).

ETN's systemic effect on MAP (Fig. 3) strongly correlates with altered microvascular reactivity: $in$ $vitro$, ETN attenuates myogenic vasoconstriction in cremaster arteries isolated from naïve mice (Fig. 4a,b). The effect is (i) dose-dependent (Fig. 4c), (ii) specifically inhibits myogenic responsiveness (that is, ETN does not affect general vasoconstriction to phenylephrine, Fig. 4d), (iii) ineffective in arteries from $Tnf^{-/-}$ mice (Fig. 4e,f) and (iv) prevented by heat denaturing the ETN (Fig. 4g,h). Further, ETN does not alter artery distensibility (%

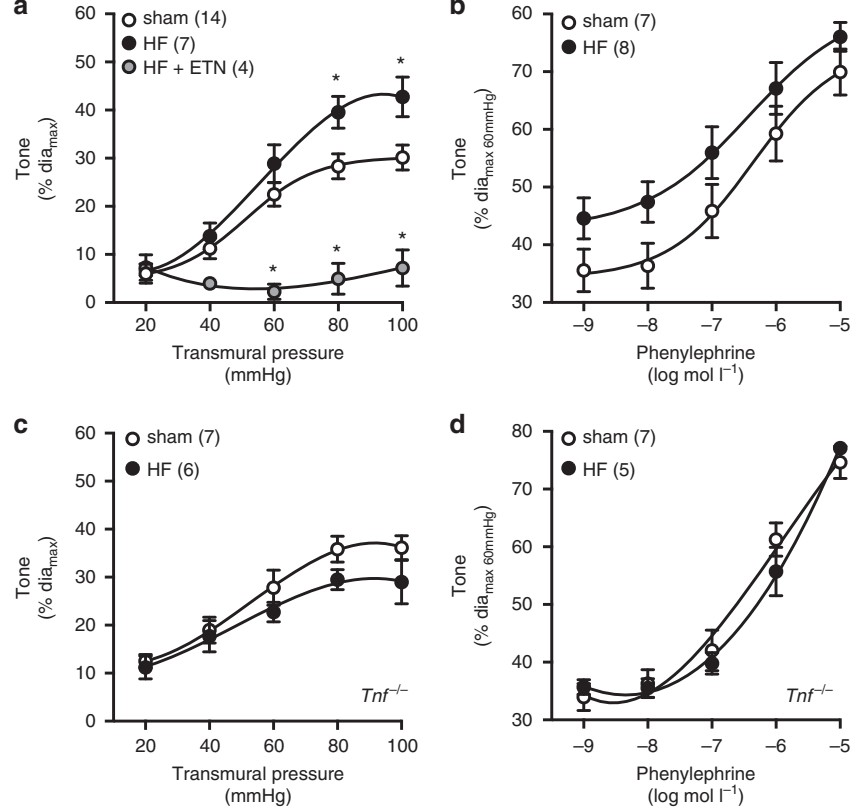

**Figure 1 | Effects of TNF disruption on myogenic responsiveness and in HF.** HF was induced by left anterior descending coronary artery ligation. Sham-operated mice served as controls. Pressure myography of cremaster skeletal muscle resistance arteries isolated from wild type and $Tnf^{-/-}$ mice. (**a**) Myogenic responsiveness and (**b**) phenylephrine-induced vasoconstriction was assessed in arteries from wild-type mice in the absence and presence of etanercept (ETN, 300 µg ml$^{-1}$ *in vitro*). (**c**) Myogenic responsiveness and (**d**) phenylephrine-induced vasoconstriction was assessed in arteries from $Tnf^{-/-}$ mice. Numbers in parentheses indicate the number of arteries in each group. All data are mean ± s.e.m. *$P < 0.05$ for one-way ANOVA and Dunnett's *post hoc* test relative to sham (**a**). An unpaired Student's *t*-test at each pressure and drug dose was nonsignificant (**b–d**). ANOVA, analysis of variance.

distension control: $10.4 \pm 0.6$ vs $10\,\mu g\,ml^{-1}$ ETN: $10.4 \pm 0.7$, $P$ = nonsignificant in a paired Student's *t*-test, $n = 5$ arteries), suggesting that TNF scavenging does not affect the serial elastic properties of the microvascular wall.

ETN disrupts several established cellular/molecular events that drive myogenic vasoconstriction[29]. Specifically, increasing transmural pressure stimulates ERK1/2 phosphorylation (Fig. 4i; uncropped western blot images provided in Supplementary Fig. 3a–h), elevates intracellular calcium (Fig. 4j), stimulates myosin light chain 20 ($MLC_{20}$) phosphorylation (Fig. 4k; uncropped western blot images provided in Supplementary Fig. 3i–n) and vasoconstriction (Fig. 4l): all of these signalling events are sensitive to ETN inhibition and therefore depend on TNF signalling; ETN does not alter agonist-stimulated vasoconstriction to phenylephrine or angiotensin II (Supplementary Fig. 4). In the context of the latter, angiotensin type I receptors have been implicated as myogenic modulators[30–32]; however, ETN's lack of effect on angiotensin II-stimulated vasoconstriction implies that angiotensin type I receptor signalling is not linked to the TNF-dependent mechanism we describe. Further, the relatively modest vasoconstriction induced by angiotensin II suggests that it is not a prominent regulator of vascular tone in the arteries studied here. Of the myogenic signalling parameters examined above, ERK1/2 phosphorylation is most upstream (that is, the most proximal to TNF)[29]: we therefore substantiated our ETN data by demonstrating that Sm-TNF-KO attenuates pressure-stimulated ERK1/2 phosphorylation (Fig. 5; uncropped western blot images provided in Supplementary Fig. 5).

We strategically selected ETN for our pharmacological intervention because it possesses a high degree of species cross-reactivity. Antibody-based anti-TNF therapeutics (for example, adalimumab) are less conducive for animal models, because their affinities for mouse TNF are many magnitudes lower than for human TNF[33]. In this context, it is unclear whether this class of anti-TNF therapeutics can be effective in the murine microcirculation. In fact, adalimumab does not lower MAP (Supplementary Fig. 6) nor does it attenuate cremaster artery myogenic vasoconstriction *in vitro* (Supplementary Fig. 6). As a corollary, adalimumab, like ETN, contains a human IgG1 Fc component; therefore, the negative adalimumab results exclude the possibility that ETN's *in vivo* effects are due to an immunological response to the human IgG1 Fc component.

**TNF regulates myogenic responsiveness across diverse species.** TNF's role as a regulator of skeletal muscle resistance artery myogenic signalling is conserved across several species, including hamsters (Fig. 4g), dogs (Fig. 6a,b), pigs (Fig. 6c,d) and, most important from a clinical perspective, humans. Myogenic responses of human thoracic wall and lumbar skeletal muscle resistance arteries are virtually abolished by ETN (Fig. 6e–h). These data substantiate the conclusion that this TNF mechanism is fundamental to the control of myogenic tone and is operational in diverse species.

**Myogenic signalling routes through membrane-bound TNF.** The observation that ETN abolishes myogenic responsiveness and

the associated molecular/cellular events (Fig. 4i–l) positions TNF as a mandatory and constitutive initiating element of myogenic signalling. However, since myogenic signalling is perpetually active, the canonical mechanism of proteolytic shedding of TNF would rapidly deplete the constitutive TNF resources and is,

therefore, intuitively unsustainable. Two lines of evidence reinforce our assertion that sTNF does not mediate myogenic responsiveness: first, inhibiting TNF shedding (TAPI-1) and thus, endogenous sTNF formation, does not affect myogenic responsiveness (Fig. 7a,b). Second, sTNF does not augment myogenic

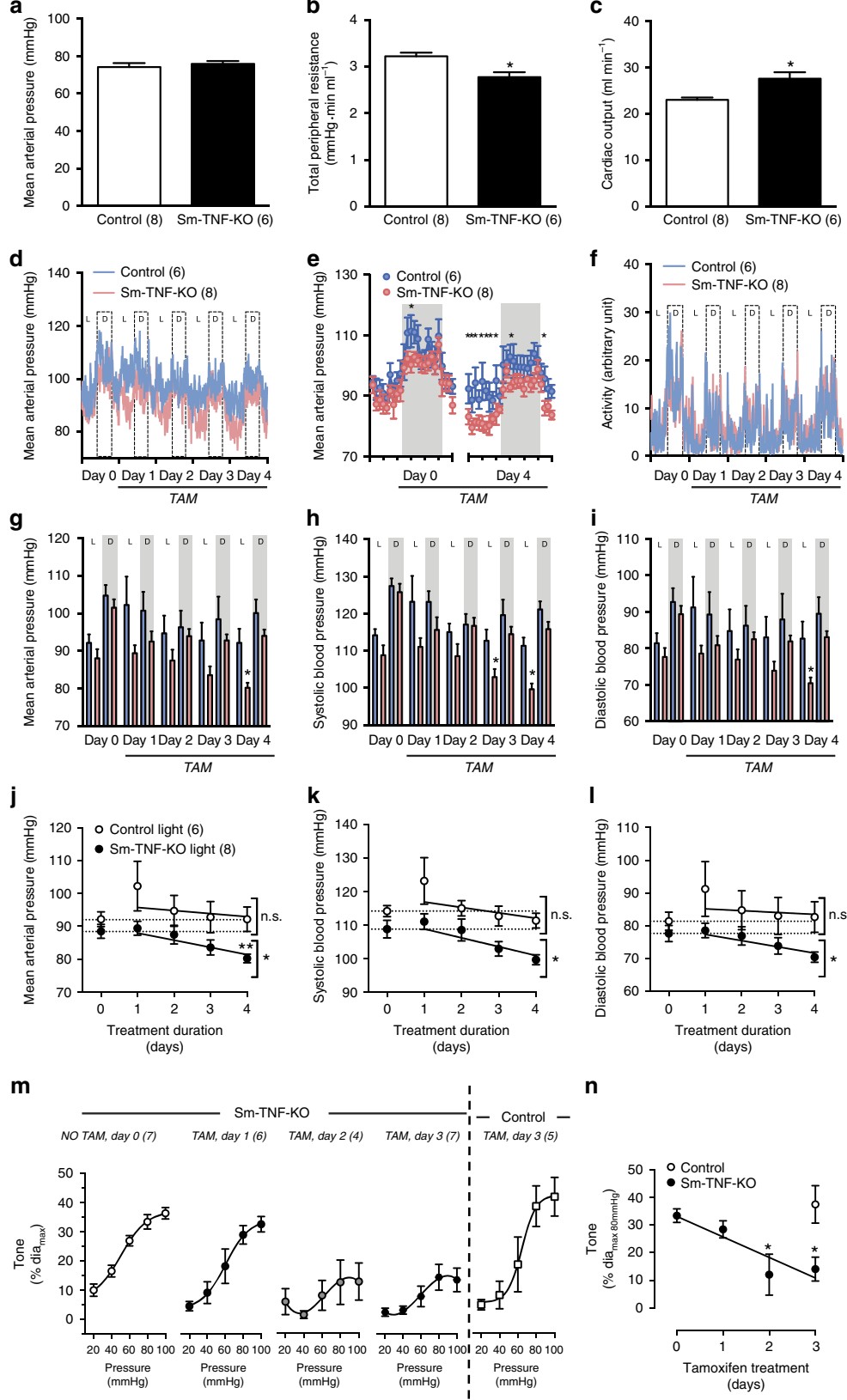

responsiveness in skeletal muscle resistance arteries (Fig. 7c,d; phenylephrine responses are normal, Supplementary Fig. 7), which would be expected if sTNF drives myogenic signalling. In fact, sTNF dose-dependently attenuates myogenic responsiveness, thereby excluding TNF receptors from transducing myogenic signals through the conventional 'forward signalling' mechanism.

By extension, the only reasonable alternative is for the myogenic signal to directly route through the membrane-bound form of TNF (mTNF), which is effectively targeted by ETN[34]. This non-canonical signalling mechanism is referred to as 'reverse signalling'[35,36]: it describes the unconventional scenario where a receptor stimulates a membrane-bound ligand and thereby initiates an 'outside-in' signal through the ligand. In skeletal muscle resistance arteries, pharmacologically activating reverse signalling with an intrinsically active TNF receptor construct (sTNFR1-Fc)[37] induces rapid vasoconstriction (Fig. 7e). Furthermore, sTNFR1-Fc augments myogenic vasoconstriction over a wide pressure range (Fig. 7f). This augmentation is absent in cremaster arteries isolated from $Tnf^{-/-}$ mice, confirming that mTNF transduces the sTNFR1-Fc-generated signal (Fig. 7g; phenylephrine responses are normal in $Tnf^{-/-}$ mice Supplementary Fig. 8). Notably, the sTNFR1-Fc effect is also absent in olfactory cerebral arteries (Supplementary Fig. 9; phenylephrine responses are normal), where sTNF enhances myogenic vasoconstriction through forward signalling[4].

Our ETN data (Fig. 4i) and previous reports[36] indicate that ERK1/2 lie downstream of mTNF-mediated reverse signalling. Accordingly, sTNFR1-Fc stimulates rapid ERK1/2 phosphorylation in mouse cremaster arteries (Fig. 7h; uncropped western blot images provided in Supplementary Fig. 10a,b). We confirmed conservation between mice and humans using more simplified models of cultured vascular smooth muscle cells (Fig. 7i,j; uncropped western blot images provided in Supplementary Fig. 10c–f). Inhibiting ERK1/2 phosphorylation (PD98059)[38] abolishes sTNFR1-Fc-induced vasoconstriction (Fig. 7k), without compromising agonist-stimulated contractility (Supplementary Fig. 11). Sphingosine kinase 1 (Sphk1) connects ERK1/2 to pro-constrictive sphingosine-1-phosphate (S1P) signalling in skeletal muscle resistance arteries[29]: thus, deleting Sphk1 disrupts this link and abolishes the sTNFR1-Fc-dependent augmentation of myogenic responsiveness (Fig. 7l; phenylephrine responses are normal in $Sphk1^{-/-}$ mice, Supplementary Fig. 12).

For mTNF to transduce a mechanical signal, it must tether to another membrane-bound protein: TNF receptors are obvious tethering partners. Because TNFR1 and TNFR2 are both expressed at the mRNA level in cremaster arteries (Supplementary Fig. 13), we selected a germline $Tnfr1/2$ double knockout model ($Tnfr1/2$ DKO) to deprive TNF from tethering. Like the germline $Tnf^{-/-}$ model, the inherited loss of TNF activity through compromised tethering should recruit alternative cellular signalling pathways for the myogenic mechanism and render ETN ineffective. As predicted, ETN fails to alter myogenic

vasoconstriction in cremaster arteries isolated from $Tnfr1/2$ DKO mice (Fig. 7m,n; phenylephrine responses are normal in $Tnfr1/2$ DKO mice, Supplementary Fig. 14). Collectively, these data support our conclusion that mTNF and TNFRs form a mechanosensitive pair that propagates a 'reverse signal' through mTNF and connects to an established ERK/Sphk1-dependent vasoconstriction mechanism (see scheme in Fig. 8).

## Discussion

This study unveils the concept that mTNF constitutively regulates skeletal muscle resistance artery myogenic vasoconstriction. At the molecular level, we identify constitutively expressed smooth muscle cell TNF as a key player in the chain of events that convert a pressure stimulus into biochemical signals and ultimately, vasoconstriction. We observe this TNF-dependent mechanism in skeletal muscle resistance arteries isolated from mice, hamsters, dogs and pigs; we also complete the first translation of this mechanism, by demonstrating its operation in two distinct human skeletal muscle resistance artery beds. This striking similarity in how different species depend on mTNF to initiate myogenic signalling suggests that this newly discovered mechanism is fundamental and emerged early in phylogenetic ancestry. TNF's new role as a mechanosensor that putatively transduces mechanical stretch into intracellular signals adds a unique element to its portfolio of biological effects and necessitates a prominent shift in how we view TNF signalling in health and disease.

Microvascular myogenic signalling is continuous[39]: it is imperative, therefore, that the elements of the signalling chain do not deplete. Constitutive proteolytic shedding[11,12] is not sustainable, effectively excluding the release of soluble TNF and/ or soluble TNF receptors as viable myogenic mechanisms. Although TNF receptors are clearly involved in myogenic signal generation, we exclude a 'forward signal' through the receptors, since soluble TNF does not mimic a pressure stimulus. There is only one sustainable configuration that does not rapidly degrade as it operates and reconciles our observations: mTNF tethers to and creates a mechanosensitive protein–protein pair with a TNF receptor that is capable of sensing (for example, via conformational changes) and transducing the pressure stimulus into a myogenic signal through mTNF (Fig. 8). This configuration is the only scenario where exogenously applied soluble TNF could simultaneously (i) disrupt the interaction between mTNF and its tethering receptor partner, without (ii) initiating a pro-constrictive forward signal through TNFRs. Although we clearly demonstrate that mTNF tethers to a TNF receptor, our experimental strategy, which utilized double TNF receptor knockout mice, does not define which TNF receptor subtypes are involved in this mechanotransduction process.

Even in the absence of structural information demonstrating a biologically relevant conformational change in response to pressure, our data permit the conclusion that mTNF is an integral

**Figure 2 | Haemodynamic and vascular responses after smooth muscle Tnf deletion.** Mice expressing inducible Cre recombinase within their smooth muscle (SMMHC-CreER$^{T2}$) and carrying either the wild-type Tnf gene (Tnf$^{wt/wt}$, control) or a floxed Tnf gene (Tnf$^{fl/fl}$) were treated with tamoxifen (TAM, 1 mg day$^{-1}$), resulting in smooth muscle cell-specific Tnf gene knockout (Sm-TNF-KO). Day 0 represents the untreated control condition (NO TAM). (**a**) MAP, (**b**) TPR and (**c**) cardiac output assessed by invasive catheterization and echocardiography. (**d**) Average raw radiotelemetric measurements of MAP in non-anaesthetized mice. Light and dark phases are indicated as 'L' and 'D', respectively. (**e**) Statistical analysis of MAP at each hour on day 0 and day 4. Grey shading indicates dark phase. (**f**) Average raw radiotelemetric measurements of ambulatory activity. Statistical analysis of (**g**) MAP, (**h**) systolic blood pressure, and (**i**) diastolic blood pressure binned over light and dark phases. White bars indicate control, black bars indicate Sm-TNF-KO, and grey shading indicates dark phase. Change in light phase measurements of (**j**) MAP, (**k**) systolic blood pressure, and (**l**) diastolic blood pressure during TAM treatment. (**m**) Pressure myography in isolated cremaster muscle resistance arteries from Sm-TNF-KO and control mice. (**n**) Statistical comparison of myogenic tone developed at 80 mm Hg. Numbers in parentheses indicate the number of mice in each group (**a–l**) or the numbers of arteries in each group (**m,n**). All data are mean ± s.e.m.; *$P < 0.05$; **$P < 0.05$ relative to untreated Sm-TNF-KO following Dunnett's *post hoc* test; n.s., nonsignificant. Student's unpaired $t$-test (**a–i**); repeated-measures one-way ANOVA and Dunnett's *post hoc* test compared with day 0 (**j–l,n**). ANOVA, analysis of variance.

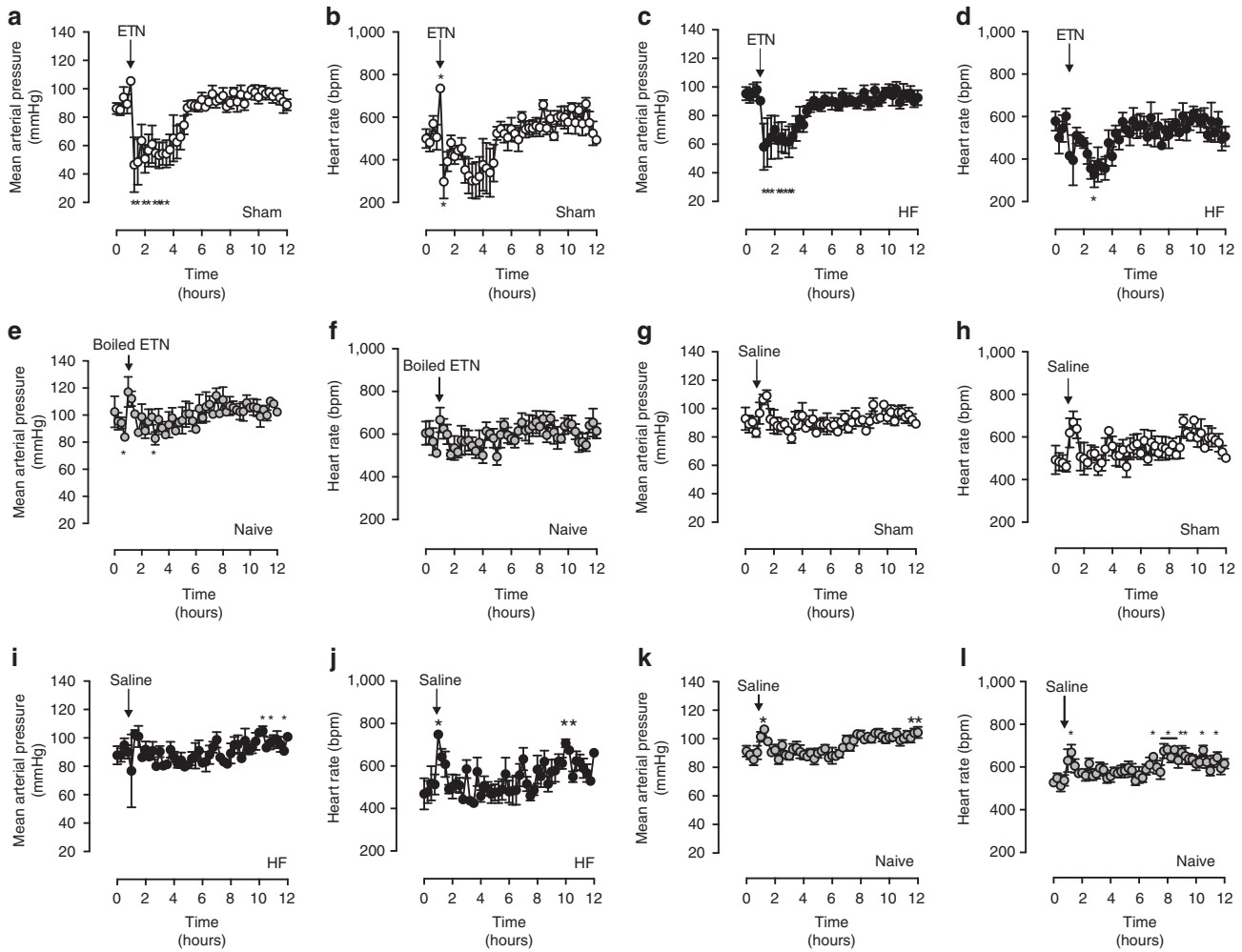

**Figure 3 | Effects of acute TNF scavenging on systemic haemodynamics.** Radiotelemetric haemodynamic measurements of sham-operated mice. (**a**) MAP and (**b**) heart rate in sham-operated mice injected with etanercept (ETN, 20 mg kg$^{-1}$ i.p., arrow, $n = 4$ sham mice). (**c**) MAP and (**d**) heart rate measurements of mice with HF induced by left anterior descending coronary artery ligation and injected with ETN (20 mg kg$^{-1}$ i.p., arrow, $n = 4$ mice). (**e**) MAP and (**f**) heart rate measurements of naive (non-operated) wild-type mice injected with boiled etanercept (boiled ETN, 20 mg kg$^{-1}$ i.p., arrow, $n = 4$ mice). (**g**) MAP and (**h**) heart rate of sham-operated mice injected with saline (100 ul, arrow, $n = 4$ mice). (**i**) MAP and (**j**) heart rate of HF mice injected with saline (100 ul, arrow, $n = 4$ mice). (**k**) MAP and (**l**) heart rate of naive (non-operated) wild-type mice injected with saline (100 ul, arrow, $n = 8$ mice). All data are mean ± s.e.m. *$P < 0.05$ for one-way repeated-measures ANOVA and Dunnett's *post hoc* test relative to 0 h. ANOVA, analysis of variance.

component of a mechanosensitive complex. This role positions mTNF upstream of canonical myogenic signals: indeed, TNF inhibition abrogates the proximal pressure-stimulated signals of ERK1/2 phosphorylation and intracellular calcium elevation[29]. sTNFR1-Fc, a ligand for mTNF[36,37], elicits ERK1/2 phosphorylation and stimulates vasoconstriction: the commonality with pressure-stimulated signals fuels our speculation that the myogenic signal routes directly through the mTNF molecule itself (that is, a 'reverse signal')[10]. The physiological functions of TNF reverse signalling are an emerging field[35–37,40]: this study presents the first evidence for this mechanism's operation in a vital microvascular function. This seminal discovery, therefore, suggests that anti-TNF therapy has an unappreciated capability to modulate vascular resistance and hence, modify haemodynamic parameters.

Remarkably, the mTNF/reverse signalling mechanism observed in skeletal muscle resistance arteries is not common to all vascular beds. Cerebral arteries are (i) insensitive to sTNFR1-Fc ligand stimulation and (ii) clearly utilize a canonical soluble TNF/TNF receptor (that is, forward signalling) mechanism to modulate myogenic reactivity[3,4,19]. Collectively, the data from these two

vascular beds suggest that mTNF reverse signalling and sTNFR/TNFR forward signalling are mutually exclusive mechanisms that do not simultaneously modulate myogenic tone within a specific vascular bed. This functional specialization necessitates a revision in how cardiovascular inflammation is managed, since anti-TNF therapeutics will have differential effects on vascular tone, depending on whether the respective vascular bed utilizes forward or reverse signalling to sustain myogenic reactivity. The immediate clinical ramification is that anti-TNF therapy must be more cautiously applied and monitored.

Anti-TNF therapy has revolutionized the management of many chronic inflammatory diseases and is, therefore, utilized in an ever-growing number of patients. Certain side effects (for example, vertigo and headaches)[41] could well result from altered systemic haemodynamic; however, the clinical trials that underpin the use of anti-TNF therapy have never been specifically designed to rigorously assess possible haemodynamic changes. Nevertheless, the BeSt clinical trial determined that anti-TNF therapy induces a significant and persistent drop in both systolic and diastolic blood pressure[42,43]. This drop in blood pressure does not associate with microvascular endothelial function

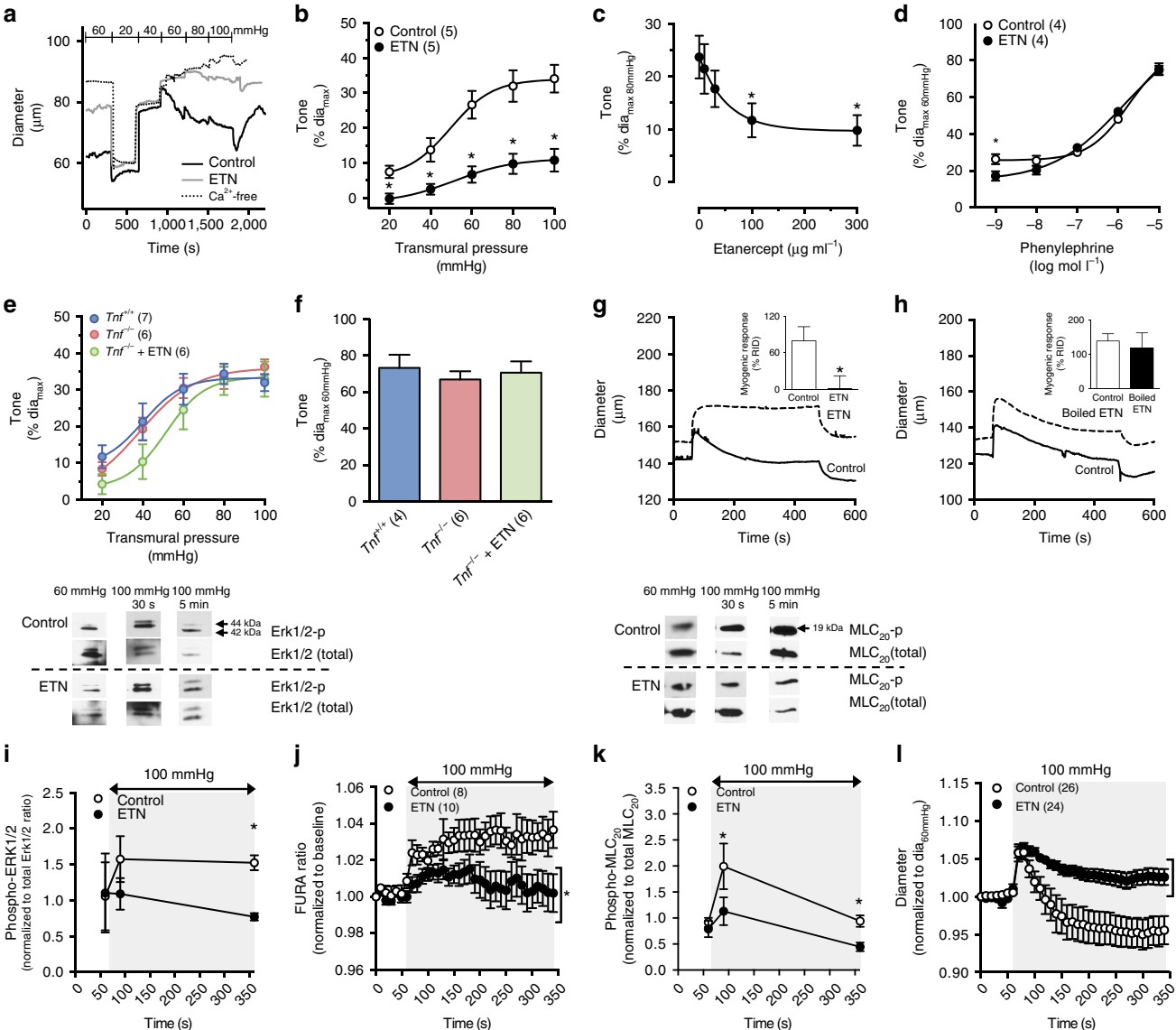

**Figure 4 | Acute TNF scavenging attenuates myogenic vasoconstriction.** (**a**) Representative traces of myogenic vasoconstriction in mouse cremaster arteries in the presence of ETN ($300 \, \mu g \, ml^{-1}$ *in vitro*) and (**b**) statistical comparison. (**c**) Dose-dependency of ETN's effect at $80 \, mm \, Hg$ ($n = 5$ arteries at 0, 7 at 10, 6 at 30, 6 at 100 and 5 at $300 \, \mu g \, ml^{-1}$). (**d**) Phenylephrine-stimulated vasoconstriction (ETN, $300 \, \mu g \, ml^{-1}$). (**e**) Myogenic and (**f**) $10 \, \mu mol \, l^{-1}$ phenylephrine-stimulated vasoconstriction in cremaster arteries from $Tnf^{+/+}$ and $Tnf^{-/-}$ with and without ETN ($300 \, \mu g \, ml^{-1}$). (**g,h**) Representative traces shown of myogenic vasoconstriction ($45–100 \, mm \, Hg$ pressure step) in hamster gracilis arteries with (**g**) native ETN ($10 \, \mu g \, ml^{-1}$; inset: comparison in $n = 6$ arteries per group) and (**h**) heat-denatured ETN ($10 \, \mu g \, ml^{-1}$; inset: comparison in $n = 6$ arteries per group). Myogenic responses displayed as reversal of initial distension (RID). (**i**) Western blot assessments of ERK1/2 phosphorylation ($60–100 \, mm \, Hg$ pressure step; control: $n = 4$ at 60, 5 at 90 and 4 at 360 s; ETN: $n = 3$ at 60, 5 at 90 and 4 at 360 s). Representative images above and uncropped images in Supplementary Fig. 3a–h. (**j**) Intracellular $Ca^{2+}$ ($60–100 \, mm \, Hg$ pressure step). (**k**) Western blot assessments of $MLC_{20}$ phosphorylation ($60–100 \, mm \, Hg$ pressure step; control: $n = 11$ at 60, 6 at 90 and 7 at 360 s; ETN: $n = 4$ at 60, 6 at 90, and 5 at 360 s). Representative images above and uncropped images in Supplementary Fig. 3i–n. (**l**) Myogenic response ($60–100 \, mm \, Hg$ pressure step; normalized to 10 s average baseline diameter; control: $62 \pm 2$, ETN: $68 \pm 2 \, \mu m$). Shading denotes $100 \, mm \, Hg$. Parentheses indicate number of arteries per group. Data are mean $\pm$ s.e.m.; *$P < 0.05$ in Student's unpaired *t*-test (**b,d,i–l**), in Dunnett's *post hoc* comparison to concentration 0 following a one-way ANOVA (**c**), and Student's paired *t*-test (**g,h**, insets). In (**e,f**), the one-way ANOVAs are nonsignificant. ANOVA, analysis of variance.

(despite transient improvement in vasodilator responses)[42] and was observed in both successful and failed interventions (as measured by disease activity scores)[43]. Taken together, these results suggest that anti-TNF therapy lowers blood pressure independent of inflammatory status by a non-endothelial mechanism. A recent report by Yoshida *et al.*[44] identified a diurnal variation in the treatment's effect on blood pressure, highlighting the need for additional clinical studies with highly controlled experimental designs.

Anti-TNF therapy can clearly worsen HF[45–48] and consequently, it is only indicated for patients with no history of HF or very specific cases of mild, well-compensated HF[47]. Because microvascular effects were never considered in the respective study designs[45,46], a potential contraindication tied to peripheral resistance modulation cannot be ascertained. Presumably, patients with normal cardiac function (for example, arthritis sufferers) tolerate the reductions in TPR resulting from anti-TNF therapy. However, HF patients have compromised cardiac

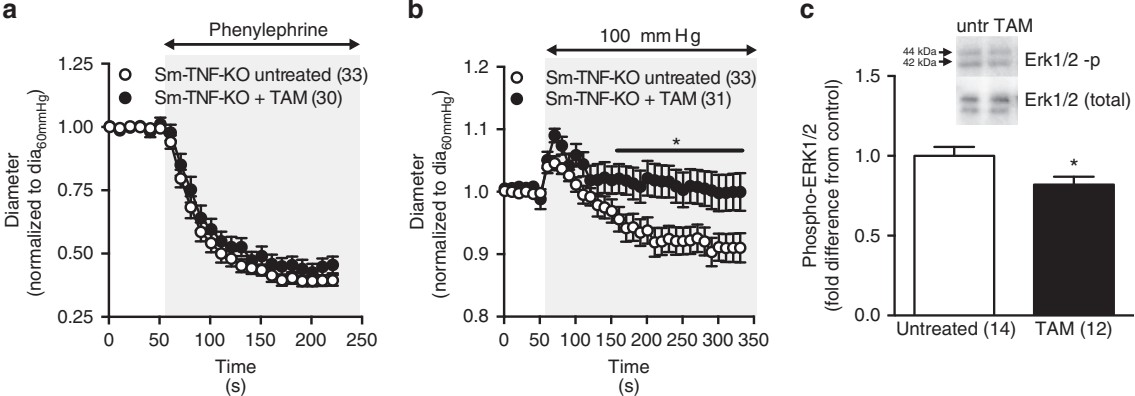

**Figure 5 | Pressure-induced ERK phosphorylation after smooth muscle *Tnf* deletion.** Cremaster muscle arteries were excised from tamoxifen treatment (TAM, 1 mg day$^{-1}$ for 3 days) in Sm-TNF-KO mice and untreated controls. (**a**) Phenylephrine-induced (10 μmol l$^{-1}$, grey shading) vasoconstriction in cremaster arteries isolated from untreated and TAM-treated Sm-TNF-KO mice. (**b**) Diameter changes in response to a single-step increase in transmural pressure (60–100 mm Hg, grey shading). (**c**) phospho-ERK1/2 levels in cremaster arteries isolated from untreated and TAM-treated Sm-TNF KO mice (normalized to total ERK1/2, expressed as fold-difference from untreated controls on same blot). Inset: representative western blot. Two arteries were pooled for each sample, and densitometry reflects 12–14 samples per group on five separate blots. Uncropped western blot images are shown in Supplementary Fig. 5. Numbers in parentheses indicate number of arteries per group (**a**,**b**) and number of replicates per group (**c**). Data are mean ± s.e.m. *$P < 0.05$ in an unpaired Student's *t*-test. ANOVA, analysis of variance.

function and depend on elevated TPR to maintain normal MAP: the unnecessary strain imposed on the heart (that is, a compensatory increase in CO) could explain why the intervention fails to deliver an improved outcome[46].

Throughout this investigation, we have pharmacologically inhibited TNF with ETN, a decoy receptor construct that exhibits strong species cross-reactivity. ETN's efficacy in our mouse models was pivotal for unmasking mTNF's haemodynamic function and the underlying molecular mechanisms. There are other pharmacological mechanisms that can inhibit TNF (for example, the antibody adalimumab); however, species cross-reactivity is problematic. For example, Food and Drug Administration-mandated pharmacological reviews indicate that adalimumab has at least 1,000-fold less affinity for murine TNF, relative to human TNF[49]; indeed, the low cross-reactivity limited the use of mouse models in the Food and Drug Administration approval process[33]. Low affinity presumably explains why adalimumab did not exert a microvascular effect in our murine models. Thus, our conclusions pertaining to anti-TNF therapy are predicated on ETN and antibody-based therapeutics inducing similar inhibitory effects on mTNF in a human setting.

In summary, the seminal discoveries presented here indicate that anti-TNF therapy has an unappreciated capability to modulate peripheral resistance and hence, haemodynamic parameters. The immediate clinical ramification is that anti-TNF therapeutics must be more cautiously applied and monitored, especially in patients with compromised cardiac function[45,46]. Our investigation unveils the concept that mTNF constitutively regulates skeletal muscle resistance artery myogenic vasoconstriction. By demonstrating that mTNF utilizes a non-canonical reverse signalling mechanism to initiate myogenic signalling, we provide evidence that TNF reverse signalling drives critical cardiovascular functions *in vivo*. The data presented herein uncovers TNF's role as a mechanosensor in smooth muscle cells and demonstrates its pivotal contribution to the transformation of transmural stretch into intracellular biochemical signals. This highly conserved function may very well extend beyond the cardiovascular system, as the ability to sense mechanical force is an essential operation found in virtually every cell.

## Methods

**Ethics approval.** All procedures conform to the Guide for the Care and Use of Laboratory Animals published by the NIH (Publication No. 85–23, revised 1996). All experimental protocols were approved by the Institutional Animal Care and Use Committee at the University of Toronto and conducted in accordance with Canadian animal protection laws. All animals were housed under a standard 14:10 h light-dark cycles with *ad libitum* access to water and chow. The use of human subjects in this study conforms to the principles outlined in the Declaration of Helsinki and was approved by the Research Ethics Board of St. Michael's Hospital, Toronto, Canada (REB #11–198). All patients provided informed written consent through Research Ethics Board-approved consent forms before study enrolment.

**Mouse models.** Wild-type (WT) male C57/BL6 mice (2–3 months) were purchased from Charles River Laboratories (Saint-Constant, Canada) and germline *Tnf* knockout mice ($Tnf^{-/-}$)[50] from Taconic Biosciences (Hudson, USA). Germline sphingosine kinase 1 knockout mice ($Sphk1^{-/-}$) were a gift from Dr. Richard L. Proia (Bethesda, MD)[51]. Germline $Tnfrsf1a^{tm1lmx}$ (p55) and $Tnfrsf1b^{tm1lmx}$ (p75) double knockout mice (Tnfr1/2 DKO) and wild-type mixed background strain controls (B6.129F2) were purchased from the Jackson laboratory (Bar Harbor, ME).

Male tamoxifen-inducible smooth muscle-specific *Tnf* knockout mice (SMMHC-*Cre*ER$^{T2}$ *Tnf$^{flox/flox}$*, abbreviated Sm-TNF-KO) were generated using a previously described breeding scheme[21], with fully characterized SMMHC-*Cre*ER$^{T2}$ and *Tnf$^{flox/flox}$* mice[22] as founders. Only male *Tnf$^{flox/flox}$* offspring express SMMHC-*Cre*ER$^{T2}$; therefore, treatment of male mice with tamoxifen (1 mg day$^{-1}$, dissolved in corn oil, intraperitoneal injection) activates the Cre recombinase and excises *Tnf* at the LoxP sites from smooth muscle cells, yielding Sm-TNF-KO[21]. Untreated SMMHC-*Cre*ER$^{T2}$ *Tnf$^{flox/flox}$* mice (that is, no TAM injection) and wild-type littermates (SMMHC-*Cre*ER$^{T2}$ *Tnf$^{wt/wt}$*, with or without TAM injection) served as controls.

Mouse skeletal muscle resistance arteries were isolated from the cremaster muscle[52]; the isolation and experimental procedures utilized 'MOPS-buffered saline' solution containing (mmol l$^{-1}$): NaCl 145, KCl 4.7, CaCl$_2$·2H$_2$O 1.5, MgSO$_4$·7H$_2$O 1.17, NaH$_2$PO$_4$·2H$_2$O 1.2, pyruvate 2.0, EDTA 0.02, (3-morpholinopropanesulfonic acid (MOPS) 3.0, and glucose 5.0. Briefly, mice were killed using an overdose of isoflurane; the testicles were exteriorized and carefully separated from attached connective tissue and the cremaster muscle was removed. Cremaster resistance arteries were carefully dissected in 4 °C MOPS under a surgical microscope (Leica MZ16) with a cold lighting source (Leica KL 1500 LCD). The resistance arteries were then cannulated and pressurized on a custom-made perfusion myography apparatus[53,54].

Mouse olfactory cerebral arteries were isolated from the brain[4]. Briefly, mice were killed using an overdose of isoflurane; the head was removed and the arteries were carefully separated from neural tissue. Olfactory cerebral arteries were dissected and cannulated in a similar manner to cremaster skeletal muscle resistance arteries.

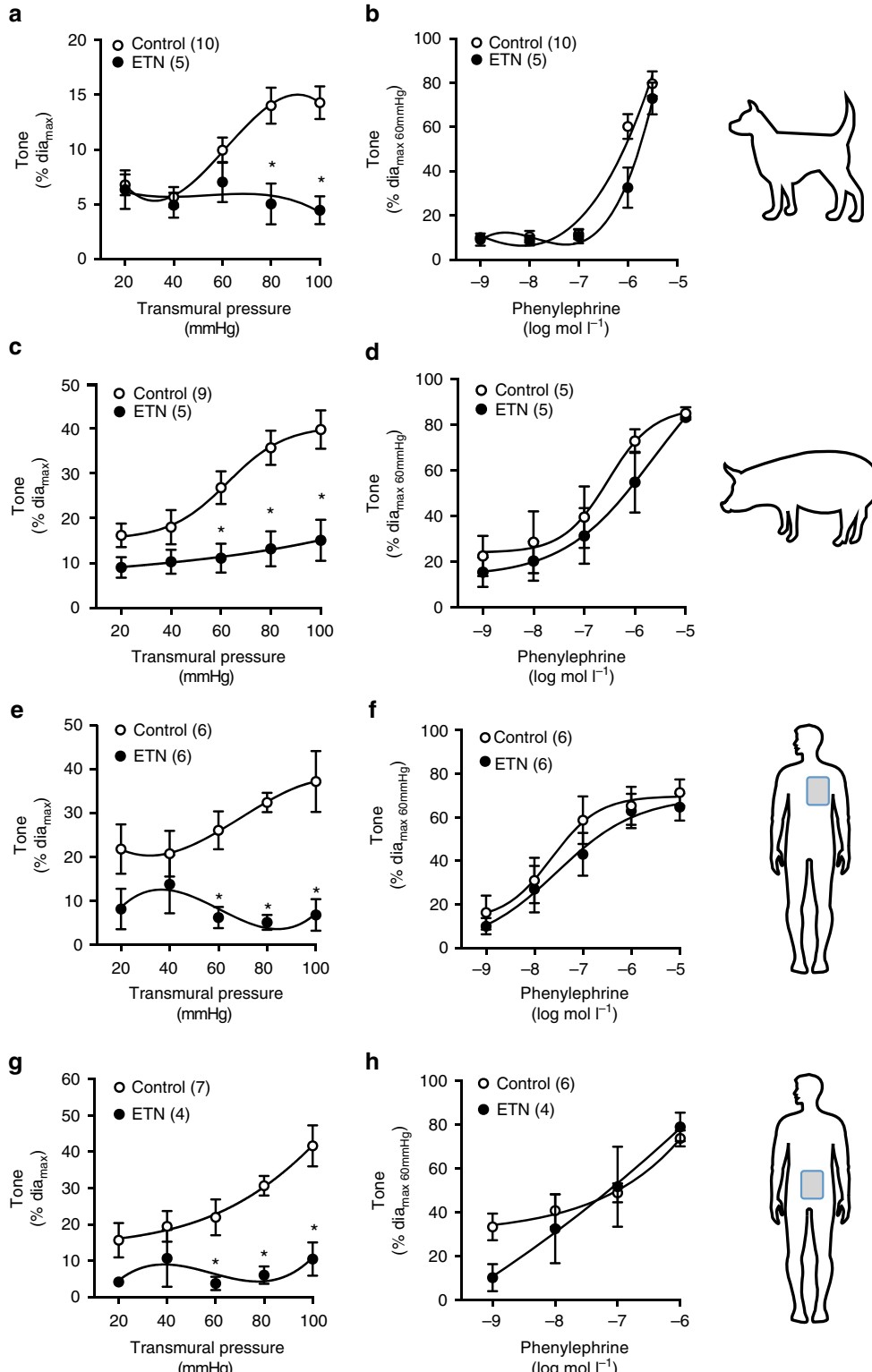

**Figure 6 | TNF scavenging in skeletal muscle arteries from animals and humans.** Myogenic and phenylephrine-stimulated vasoconstriction following treatment with etanercept (ETN, 300 μg ml$^{-1}$ *in vitro*). (**a,b**) Dog gracilis skeletal muscle resistance arteries. (**c,d**) Pig gracilis skeletal muscle resistance arteries. (**e,f**) Human thoracic wall skeletal muscle resistance arteries. (**g,h**) Human lumbar skeletal muscle resistance arteries. Numbers in parentheses indicate arteries per group. All data are mean ± s.e.m. *$P < 0.05$ for unpaired Student's *t*-test comparison at each pressure.

**Non-murine animal models.** Female Golden Syrian hamsters (2–3 months of age; Charles River Laboratories) were euthanized by isoflurane inhalation and cervical dislocation. Hamster gracilis arteries were rapidly dissected from surrounding skeletal muscle tissue, cannulated and cultured for 24 h under constant flow conditions (1 ml h$^{-1}$) (ref. 29). This procedure permits hamster gracilis arteries to recover from the surgical removal of surrounding connective tissue and regain robust responses. Before functional assessment, arteries were washed in MOPS-buffered saline.

Tissue samples from healthy adult mongrel dogs (1–3 years old) and healthy Yorkshire pigs (25–35 kg) were generously donated by Dr Paul Dorian (Keenan Research Centre for Biomedical Science at St. Michael's Hospital, Toronto, Ontario, Canada). Dogs and pigs were anaesthetized with inhaled

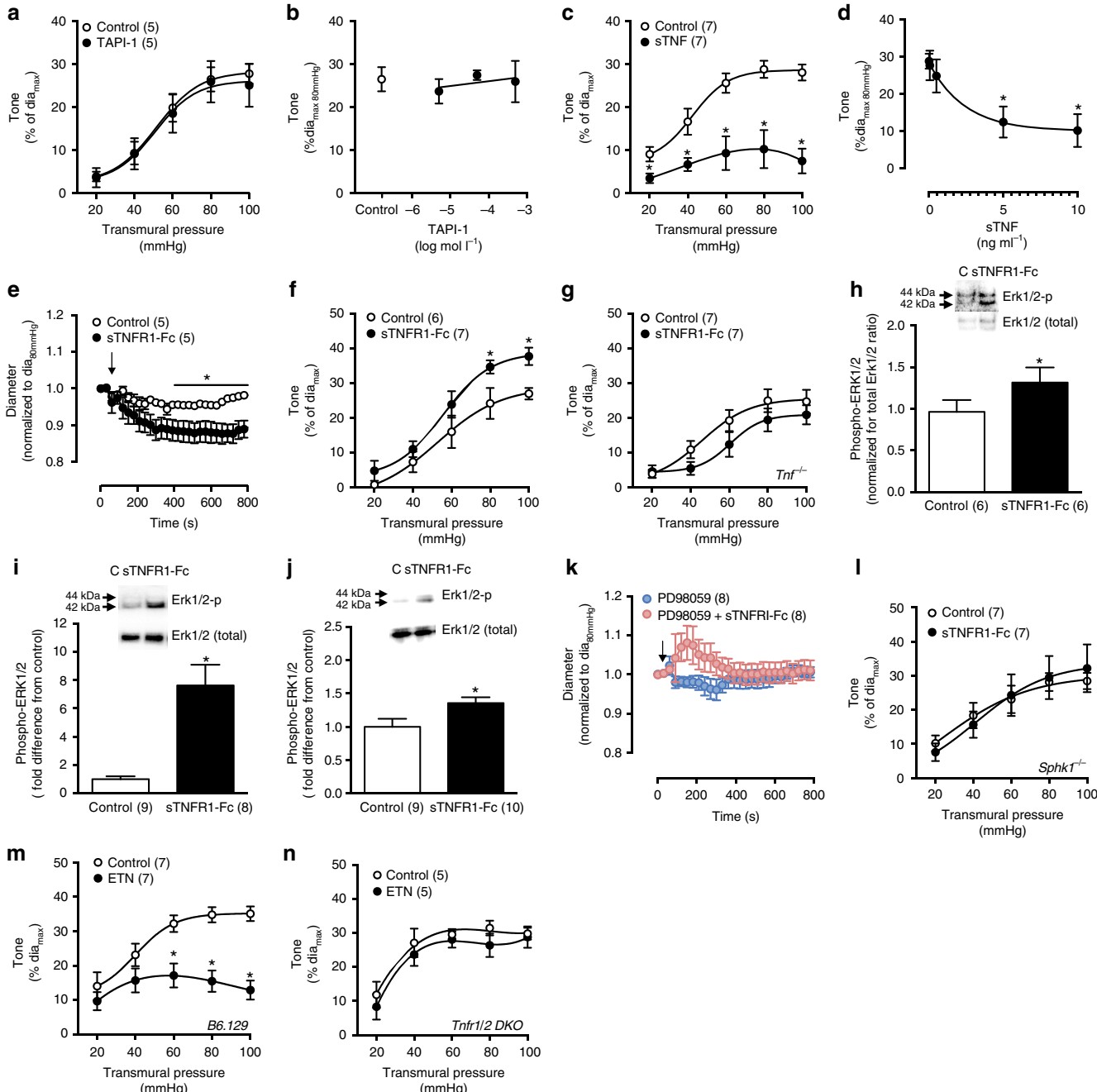

**Figure 7 | TNF reverse signalling in the myogenic response. (a)** Myogenic responses in the presence of the TNF converting enzyme inhibitor TAPI-1 (500 μmol l$^{-1}$ *in vitro*) and **(b)** dose-dependency assessment at 80 mm Hg (control: $n = 5$ arteries; 5 at $-5.3$, 4 at $-4.3$, 5 at $-3.3$ log mol l$^{-1}$ TAPI-1). **(c)** Myogenic responses in the presence of recombinant soluble TNF (sTNF, 10 ng ml$^{-1}$ *in vitro*) and **(d)** dose-dependency assessment at 80 mm Hg ($n = 7$ arteries at 0, 5 at 0.05, 7 at 0.5, 6 at 5 and 6 at 10 ng ml$^{-1}$ sTNF). **(e)** Wild-type cremaster artery diameter measurements at 80 mm Hg transmural pressure following application of an intrinsically active soluble TNFR1 fragment (sTNFR1-Fc; 100 ng ml$^{-1}$, arrow); data are normalized to baseline diameter (10 s average control: 51 ± 5, sTNFR1-Fc: 51 ± 5). **(f,g)** Myogenic responses conducted in the presence of sTNFR1-Fc *in vitro* (100 ng ml$^{-1}$) in **(f)** wild-type and **(g)** *Tnf$^{-/-}$* cremaster arteries. **(h–j)** Representative western blot assessments and statistical analysis of ERK1/2 phosphorylation under control condition (C) and following sTNFR1-Fc (100 ng ml$^{-1}$) application to **(h)** cremaster arteries at 80 mm Hg transmural pressure, **(i)** cultured mouse mesenteric vascular smooth muscle cells, and **(j)** cultured human coronary artery smooth muscle cells. Uncropped western blot images are shown in Supplementary Fig. 10. **(k)** Wild-type cremaster artery diameter measurements at 80 mm Hg transmural pressure following application of sTNFR1-Fc (100 ng ml$^{-1}$, arrow) in the presence of 10 μmol l$^{-1}$ PD98059; data are normalized to baseline diameter (10 s average control: 51 ± 4, sTNFR1-Fc: 51 ± 4). **(l)** Myogenic responses in the presence of sTNFR1-Fc *in vitro* (100 ng ml$^{-1}$) in cremaster arteries isolated from *Sphk1$^{-/-}$* mice. **(m,n)** Myogenic responses in cremaster arteries isolated from **(m)** B6.129 wild-type controls and **(n)** *Tnfr1/2 DKO* mice in the presence and absence of etanercept (ETN, 300 μg ml$^{-1}$ *in vitro*). Numbers in parentheses indicate the number of arteries per experimental group (**a,c,e–g,k–n**) or the number of replicates per experimental group (**h–j**) All data are mean ± s.e.m. *$P < 0.05$ for unpaired Student's *t*-test (**a,c,e–g,i–n**), paired Student's *t*-test (**h**, paired within blot), and one-way ANOVA with Dunnett's *post hoc* comparison to concentration 0 (**b,d**). ANOVA, analysis of variance.

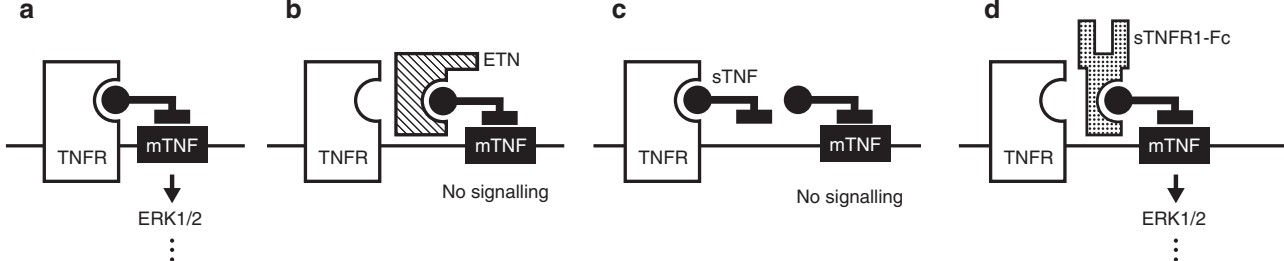

**Figure 8 | Proposed scheme for TNF as a mechanosensor. (a)** In the control setting, tethering between mTNF and TNF receptors (TNFR) provides the structural basis for strain to elicit a reverse signal through mTNF that links to ERK1/2. **(b)** Etanercept (ETN) which lacks intrinsic activity, interferes with mTNF/TNFR tethering and hence, reverse signalling. **(c)** Soluble TNF (sTNF) competes with mTNF for TNFR binding, which also interferes with mTNF/TNFR tethering and hence, reverse signalling. **(d)** The soluble TNFR1 fragment (sTNFR1-Fc) interferes with mTNF/TNFR tethering, but stimulates a reverse signal through its intrinsic activity.

isoflurane and were killed by removal of the heart (dogs) or ventricular fibrillation (pigs). Once the animals had been killed, a section of gracilis muscle was removed; resistance arteries were dissected from surrounding skeletal muscle tissue, cannulated and functionally assessed by perfusion myography.

**Human tissue samples.** With informed consent (REB #11–198), human skeletal muscle biopsies were collected from (i) cardiac patients undergoing elective coronary artery bypass graft (CABG) surgery for coronary artery disease and occlusion and (ii) patients undergoing lumbar laminectomy for vertebral disk herniation and spinal stenosis. All patients had normal ejection fraction and were not treated with anti-TNF therapeutics (Supplementary Table 3). Surgeons directly provided a small piece of human intrathoracic fascia-muscle (CABG surgery) or paraspinous muscle (lumbar) to research staff inside the operating room; these biopsies (1–2 cm$^2$) were collected without cautery, and gradually cooled in buffer to 4 °C before the resistance arteries were dissected from the surrounding skeletal muscle tissue.

**Invasive haemodynamic measurement.** Systemic blood pressure was assessed by invasive catheterization[52]. Briefly, mice were anaesthetized (1.7% isoflurane), the carotid artery was exposed and a catheter (1.4 F sensor 2.0 F catheter; Millar Inc.; Houston, USA) was extended into the thoracic aorta and secured using a suture for blood pressure measurement. The catheter was then extended into the left ventricle and maximum developed pressure (dP/dt max) and minimum developed pressure (dP/dt min) were measured and analysed using GraphPad Prism (GraphPad Software Inc; La Jolla, USA). Mice were subsequently killed and the heart was removed and weighed.

**Telemetric haemodynamic measurements.** Continuous haemodynamics and mouse activity were gathered using blood pressure telemetry (PhysioTel PA-C10 Pressure Transmitter for Mice and Data Exchange Matrix; Data Sciences International, St. Paul, USA). Briefly, mice were anaesthetized with inhaled isoflurane and the carotid arteries were carefully exposed. A gel filled catheter, attached to the blood pressure transducer (which communicates wirelessly with the receiver), was inserted into the right common carotid artery and secured with a silk suture. The blood pressure transducer was implanted in a subcutaneous pouch on the chest. Mice recovered over 7 days and, after 2–4 days of baseline recording, mice were injected with tamoxifen (1 mg day$^{-1}$ intraperitoneal between 4–5 h after light-on on consecutive days) and haemodynamic parameters were recorded throughout injections and in the follow-up period (10 s sample every 10 min continuously).

The standard 14:10 light:dark cycle had 'lights on' at 06:00 and 'lights off' at 20:00. To assess the diurnal variation on haemodynamic measures, values were obtained over a 3 h period during the light phase (13:00 to 16:00, inactive) and a 3 h period during the dark phase (01:00 to 04:00 active).

**Echocardiography.** Mice were anaesthetized (1.5–1.7% isofluorane) and were placed in a supine position on a heating pad and temperature was monitored with a rectal probe (THM 150, Indus Instruments). Chest fur was removed and acoustic gel was applied to the chest wall (Aquasonic 100; Parker Laboratories, Fairfield, USA). A Vevo 770 ultrasound system (VisualSonics Inc.; Toronto, Canada) was used and a transthoracic M-mode ultrasound was taken using an ultrasonic linear transducer scanning head set to 30 MHz. Using the long-axis view, left ventricular-end-diastolic volume (LVEDV), left ventricular-end-systolic volume (LVESV), diastolic diameter (Dd), and systolic diameter (Sd) were obtained. Stroke volume (SV) was calculated as SV = LVEDV-LVESV. Using simultaneous heart rate (HR), CO was calculated as CO = HR × SV. Ejection fraction (EF, in %) was calculated as EF = SV/LVEDV.

**Induction of murine heart failure.** WT and $Tnf^{-/-}$ mice underwent surgery to induce a myocardial infarction (MI) by ligation of the left anterior descending coronary artery (LAD)[19,55]. Briefly, mice were anaesthetized with isoflurane, the thorax and pericardium were opened and a needle with 7-0 silk suture was used to ligate the LAD. The pericardium and thorax were closed and the animal recovered. Mice that underwent sham procedure (chest opened, needle advanced into myocardium) served as controls. Mice were kept for 8–10 weeks, a time point when postoperative pulmonary oedema and increased heart weight indicated that HF was fully established[52]. Subsets of sham and HF mice were implanted with telemetric blood pressures probes at 6 weeks post-op. These instrumented mice received intraperitoneal injections with the TNF scavenging drug Etanercept (ETN; Enbrel, Amgen Inc., 20 mg kg$^{-1}$), boiled ETN (20 mg kg$^{-1}$), Adalimumab (Humira, AbbVie Corp., 20 mg kg$^{-1}$), or equivalent volumes of saline and blood pressure was continuously monitored. A subset of HF mice was euthanized for pressure myography assessments.

**Pressure myography.** Following cannulation, myography units were transferred to a heating plate mounted on an inverted microscope (Nikon Eclipse TE2000-U); vessels were heated to 37 °C and stretched to in vivo lengths. Vessel viability was assessed using 0.3 µmol l$^{-1}$ norepinephrine (in hamsters), or 10 µmol l$^{-1}$ phenylephrine (in mice, dogs, pigs and humans), or 60 mmol l$^{-1}$ KCl (humans). Only vessels that showed robust constriction (>50% tone) underwent further testing.

Pressure myography experiments were conducted in the presence of various substances including: adalimumab (a fully humanized anti-TNF antibody, 300 µg ml$^{-1}$ in vitro, 20 mg kg$^{-1}$ in vivo), etanercept (ETN, a fusion protein combining the two copies of the ligand binding domain of TNF receptor 2 (TNFR2) and the F$_c$ component of the human IgG$_1$, 10–300 µg ml$^{-1}$ in vitro, 20 mg kg$^{-1}$ in vivo), the TNF-cleaving enzyme inhibitor TAPI-1 (Peptides International, 5–500 µmol l$^{-1}$ in vitro), soluble recombinant human TNF (sTNF, Sigma-Aldrich, 0.05–10 ng ml$^{-1}$ in vitro), the MEKK1 inhibitor PD98059 (Cayman Chemical, 10 µmol l$^{-1}$ in vitro), and recombinant mouse TNF R1/TNFRAF1A Fc Chimera (sTNFR1-Fc, carrier free, R&D Systems, 100 ng ml$^{-1}$ in vitro). All substances were dissolved in MOPS buffer and incubated for 30 min at 37 °C before assessing myogenic responsiveness.

To assess myogenic responsiveness, arteries were exposed to step-wise increases in transmural pressure (20–100 mm Hg in 20 mm Hg increments for 5 min in mouse cremaster, dog, pig and human resistance arteries, 20–80 mm Hg in mouse olfactory cerebral arteries). In experiments in which drugs were serially tested (that is, first control response followed by second response with drug), responses were compared with separate experiments in which serial responses were conducted only in MOPS buffer (that is, time control). At the completion of each experiment, vessels were incubated in Ca$^{2+}$-free MOPS-buffered saline solution ((mmol l$^{-1}$): NaCl 147, KCl 4.7, MgSO$_4$ · 7H$_2$O 1.17, NaH$_2$PO$_4$ · 2H$_2$O 1.2, pyruvate 2.0, EDTA 2.00, MOPS 3.0 and glucose 5.0). Following incubation with Ca$^{2+}$-free MOPS buffer, vessels underwent a step-wise increase in transmural pressure under Ca$^{2+}$-free conditions and passive diameter (dia$_{max}$) was recorded at each pressure. Myogenic tone (%) was calculated as: myogenic tone (%) = (dia$_{max}$–dia$_{response}$)/dia$_{max}$ × 100, where dia$_{response}$ is the diameter at a given pressure. For agonist-induced vasoconstriction, responses were compared with passive diameter under Ca$^{2+}$-free conditions at dia$_{60mm Hg}$ (diamax$_{60mm Hg}$) for all vascular beds. Agonist-induced responses were calculated as: tone (%) = dia$_{max 60mm Hg}$–dia$_{response}$)/dia$_{max 60mm Hg}$ × 100, where dia$_{response}$ is the diameter at a given concentration of the drug.

To assess the myogenic response in hamster arteries, transmural pressure was increased from 45 to 100 mm Hg in a single step[54,56]. Vessel diameter was measured at each pressure step once steady state was achieved (7 min; dia$_{response}$). Myogenic responsiveness was expressed as reversal of initial distension (% RID = (dia$_{100mmHg}$–dia$_{response}$)/(dia$_{100mmHg}$–dia$_{45mmHg}$) × 100).

**Calcium measurements.** Intracellular $[Ca^{2+}]$ ratio was measured by incubating pressurized mouse cremaster arteries with the calcium-sensitive Fura 2 ratiometric dye ($2\,\mu mol\,l^{-1}$ Fura 2-AM; incubated 1 h at 25 °C and 1 h at 37 °C)[29]. Of note, changes in Fura 2 ratios represent different increments of $Ca^{2+}$, not exact concentrations. Briefly, excitation of $Ca^{2+}$-bound Fura 2 (340 nm) or $Ca^{2+}$-unbound Fura 2 (380 nm) was switched $2\times$ per second with a high-speed random access monochromator (Photon Technology International). The resulting 510 nm emission was captured by a photomultiplier for quantification and simultaneous vessel diameter measurement. The Fura 2 ratio (340/380) indicates intracellular $Ca^{2+}$ concentration and was expressed as a fraction of the baseline ratio of each vessel.

**Western blotting.** Cremaster arteries from wild-type and Sm-TNF-KO mice (untreated and treated with tamoxifen for 3 days) were isolated and cannulated; only viable vessels were utilized in western blots. For sTNFR1-Fc experiments, vessels were set at 80 mm Hg transmural pressure and were incubated in MOPS or sTNFR1-Fc ($100\,ng\,ml^{-1}$) for 5 min vessels were immediately submerged into 4 °C protein extraction buffer containing protease and phosphatase inhibitors (Roche Scientific).

For ETN experiments, viable wild-type mouse cremaster arteries were incubated at 60 mm Hg transmural pressure in MOPS or treatment group (that is, ETN, 30 min). Vessels that were maintained at 60 mm Hg served as controls. For Sm-TNF-KO experiments, viable cremaster arteries from untreated or TAM-treated mice were incubated at 60 mm Hg in MOPS. Following incubation, vessels underwent a single pressure step from 60 to 100 mm Hg and responses were recorded. After either 30 s or 5 min at 100 mm Hg, vessels were immediately submerged into 4 °C protein extraction buffer (1% sodium dodecyl sulfate, 10% glycerol, $20\,mmol\,l^{-1}$ Tris pH 6.8, $1\,mmol\,l^{-1}$ EDTA, $0.7\,mol\,l^{-1}$ β-mercaptoethanol) containing protease inhibitor (Roche Scientific) and okadaic acid (1:20).

Proteins from 2 to 4 pooled cremaster arteries were extracted by mechanical homogenizing and repeated freeze-thaw cycles. Proteins were separated by electrophoresis on a standard SDS-polyacrylamide gel electrophoresis gel (9–15% bis-acrylamide).

ETN experiments used a standard western blot method[29]. Following transfer, PVDF membranes were blocked (5% skim milk; 1 h) and incubated overnight with rabbit anti-phospho-p44/42 MAPK (ERK1/2) antibody (1:1,000; Cat #4377S, Cell Signaling Technology) or mouse anti-phospho-MLC$_{20}$ antibody (1:500, Cat #3675, Cell Signaling Technology), and incubated with either horseradish peroxidase-linked anti-rabbit IgG (1:10,000; Millipore) or horseradish peroxidase-linked anti-mouse IgG (1:10,000; Millipore). A standard chemiluminescence procedure was used to expose X-ray film. Following assessment of phosphorylated proteins, membranes were blocked (5% skim milk) and re-probed overnight with mouse anti-p44/42 MAPK (ERK1/2) antibody (1:1,000; Cat #9107S, Cell Signaling Technology) or rabbit anti-MLC$_{20}$ antibody (1:1,000, Cat #3672, Cell Signaling Technology) and incubated with respective horseradish peroxidase-linked secondary antibodies. Developed Kodak photographic films were scanned into a personal computer and densitometry was quantified using image analysis software (Image J, NIH) and expressed as % phosphorylated protein/total protein.

Western blots involving sTNFR1-Fc in wild-type mouse vessels, and all Sm-TNF-KO experiments, used a modified method that improved sensitivity and thereby reduced tissue requirements. Following transfer, PVDF membranes were fixed in gluteraldehyde solution (0.5% in PBS, 45 min) and blocked (5% milk; 1 h). A 3-step Western blotting method (biotin/streptavidin) was then used[57]. Briefly, PVDF membranes were incubated overnight with rabbit anti-phospho-p44/42 MAPK (ERK1/2) antibody (1:1,000; Cat #4377S, Cell Signaling Technology) or rabbit anti-p44/42 MAPK (ERK1/2) antibody (1:1,000; Cat #4695S, Cell Signaling Technology), incubated with donkey anti-rabbit biotin IgG (1:20,000 Millipore), and then high sensitivity streptavidin-HRP (1:500, Pierce). A standard chemiluminescence procedure was used to expose membranes. Images were captured using a digital camera system (Chemidoc Touch; Bio-Rad Laboratories, Hercules, USA) and quantified using image analysis software (Bio-Rad Image Lab).

**Reverse transcription—polymerase chain reaction.** Cremaster resistance artery RNA was isolated with 'Total RNA Purification Micro' spin columns (Norgen Biotek), as directed. Briefly, cremaster arteries were homogenized in the kit-provided lysis buffer using conical glass grinders; the ground homogenate was then subjected to proteinase K ($450\,\mu g\,ml^{-1}$ final concentration) digestion for 20 min at 55 °C. The resulting homogenate was loaded onto the spin column and washed once with the kit-provided buffer. DNase I (0.25 Kunitz units) was then applied to the column for 20 min to remove contaminating genomic DNA. The column was then washed twice and the RNA then eluted in distilled water.

Creamster artery RNA was converted to cDNA using a 'Superscript III' reverse transcription (RT) kit (Invitrogen Life Technologies). Briefly, before starting the RT reaction, the isolated RNA was combined with deoxynucleotide triphosphates (dNTPs) and random heximers, heated to 65 °C for 5 min and then chilled on ice for 2 min. The remaining components required for the RT reaction were then added: each RT reaction ($40\,\mu l$ volume) contained $8\,\mu l$ kit-provided $5\times$ buffer solution, $0.5\,mmol\,l^{-1}$ dNTPs (Genedirex Inc.), 100 ng random heximers (Invitrogen Life Technologies), $5\,\mu mol\,l^{-1}$ dithiothreitol, 400 U Superscript III

reverse transcriptase (Invitrogen Life Technologies) and 80 U RNaseOut recombinant ribonuclease inhibitor (Invitrogen Life Technologies). Each RT reaction was incubated at 25 °C for 5 min and then 50 °C for 1 h; the Superscript enzyme was then inactivated by heating to 70 °C for 15 min. Immediately following the RT reaction, RNA was removed by adding 2.5 U RNase H (New England Biolabs) and incubating the reaction at 37 °C for 20 min; the RNAse enzyme was subsequently inactivated by heating to 65 °C for 15 min.

Target genes were amplified with an i-Taq DNA polymerase kit (FroggaBio). Each reaction ($25\,\mu l$ volume) contained: $2.5\,\mu l$ kit-provided $10\times$ buffer solution, $0.8\,mmol\,l^{-1}$ dNTPs, $400\,nmol\,l^{-1}$ forward and reverse primers and 1.25 U iTaq DNA polymerase. Primer sequences for each target were (5′–3′): TNF forward CCAGTGTGGGAAGCTGTCTT; TNF reverse AAGCAAAAGAGGAGGCAACA; TNFR1 forward CAGTCTGCAGGGAGTGTGAA; TNFR1 reverse TCAGCTT GGCAAGGAGAGAT; TNFR2 forward CAGGTTGTCTTGACACCCTAC; and TNFR2 reverse GCACAGCACATCTGAGCCT. Each reaction received $1\,\mu l$ cDNA; negative controls received water instead of cDNA. The PCR amplification consisted of 5 min denaturation at 95 °C, followed by 35 cycles of amplification (60 s at 95 °C + 30 s at 60 °C + 30 s at 72 °C). The PCR products were separated on a 10% polyacrylamide gel, stained with SyberSafe DNA gel stain (Invitrogen Life Technologies) and imaged on a BioRad ChemicDoc System. Expected amplicon sizes for the targets are: TNF 100 base pairs (bp); TNFR1 124 bp; and TNFR2 92 bp.

**Genotyping for TNF flox product.** Tissue was isolated from tamoxifen-treated SM-TNFα-KO and control mice. Samples were kept at $-80$ °C to maintain tissue integrity. On the day of genotyping, a small piece of each sample was isolated on ice. DNA was isolated using the Sigma REDExtract-N-Amp Tissue PCR kit according to the manufacturer's instructions. The PCR was prepared using the REDExtract-N-Amp PCR mix. All primer stocks were $100\,pmol\,\mu l^{-1}$ and diluted into 'genomix' as described. Each reaction received $1\,\mu l$ of 'genomix' per reaction. For each sample, flox, Cre and the presence of the tamoxifen-induced knockout allele were assessed using two master mixes: Cre Genomix: $50\,\mu l$ SMWT1 + $25\,\mu l$ SMWT2 + $50\,\mu l$ Cre(phCREAS1) + $125\,\mu l$ $H_2O$, SMWT1: 5′-TGACCCCATCTCT TCACTCC-3′; SMWT2: 5′-AACTCCACGACCACCTCATC-3′, Cre(phCREAS1): 5′-AGTCCCTCACATCCTCAGGTT-3′;

TNF(Flox) Genomix: $25\,\mu l$ TNF(Flox) Fwd + $25\,\mu l$ TNF(Flox) Rev + $200\,\mu l$ $H_2O$, TNF(Flox) Fwd: 5′-TGAGTCTGTCTTAACTAACC-3′, TNF(Flox) REV: 5′-CCCTTCATTCTCAAGGCACA-3′.

TNF(TAM KO) Genomix: $25\,\mu l$ TNF(Flox) Fwd + $25\,\mu l$ TNF(Flox) KO49 + $200\,\mu l$ $H_2O$, TNF(Flox) KO49 5′-CTCTTAAGACCCACTTGCTC-3′.

The PCR cycle parameters were 95 °C for 5 min and 40 cycles of 95 °C for 45 s (denaturation), 58 °C for 1 min (annealing), and 72 °C for 1 min (extension). These cycles were followed by 72 °C for 7 min (to extend all unfinished product). A 2.5% agarose gel was made containing $1.75\,\mu l$ of RedSafe (Frogga cat #21141) to separate the amplicons. The incubated PCR mix was directly loaded onto the gel with the ladder (GeneDirex DM003-R500, $7\,\mu l$ 100–3,000 bp). After $\sim40$ min of electrophoresis (120 V) the gel was digitally imaged with a Kodak 4000 mm Pro imaging station.

**Cell culture.** Mouse vascular smooth muscle cells were isolated from mesenteric arteries[19]. Briefly, mesenteric arteries were isolated and digested with a cocktail of trypsin, collagenase and elastase. Cells were washed with phosphate-buffered saline and grown in Dulbecco's modified Eagle's medium containing 10% fetal bovine serum and 1% penicillin-streptomycin. Mouse vascular smooth muscle cells were authenticated as positive for alpha actin, myocardin and MyoD, and negative for endothelial nitric oxide synthase. These cells were not tested for mycoplasma. Human coronary artery smooth muscle cells (Lonza CC-2583) were grown in smooth muscle cell basal medium (SmBM, Lonza) containing growth factors, cytokines and supplements (SingleQuots Kit, Lonza). Human cells are positive for alpha smooth muscle actin, negative for von Willebrand Factor VIII, and negative for mycoplasma. Cells were grown on 12 well tissue culture plates and were maintained at 37 °C with 5% $CO_2$. Once cells reached 80–90% confluence, they were washed with $Ca^{2+}$-containing PBS and treated with sTNFR1-Fc ($100\,ng\,ml^{-1}$, 5 min) or PBS alone. Cells were rapidly lysed with 4 °C protein extraction buffer containing protease and phosphatase inhibitors (Roche Scientific) and scraped from the culture dish. Samples were then sonicated, boiled (95 °C 10 min), spun in a centrifuge (13,500g, 10 min) and supernatant was removed and underwent the three-step western blotting protocol described above.

**General experimental design and statistical analyses.** Sample size was based on previous publications[19,29,52]; power calculations, therefore, were not necessary. Animals were arbitrarily allocated to the various groups, without the use of an explicit randomization procedure. Analyses did not require blinding and were not, therefore, performed under blinded conditions. In this report, no data points, samples, or mice were excluded from analysis. All statistical analyses are deemed appropriate based on our previous publications[29,52].

All data are expressed as means ± s.e.m.; n values represent the number of mice or arteries, as specified. Variance is similar between comparison groups. All data were analysed with a Student's unpaired or paired t-test or a one-way ANOVA,

followed by Dunnett's *post hoc* tests. Differences were considered significant at $P < 0.05$.

**Data availability.** The data that support the findings of this study are available within the article and its supplementary information files. All data can be made available by the corresponding author upon request.

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

## Acknowledgements

We gratefully acknowledge the following organizations for providing Operating, Salary and Infrastructure Funding: Ontario Graduate Scholarship—Science and Technology (J.T.K.), Heart and Stroke Foundation of Ontario—Queen Elizabeth II Scholarship (J.T.K.), Heart and Stroke/Richard Lewar Centre of Excellence (J.T.K.), Natural Sciences and Engineering Research Council (J.T.K.), Operating and Infrastructure Grants from the Canadian Institutes of Health Research (S.-S.B., MOP-84402 and MOP-136957; PHB, MOP-119339), the Russian Science Foundation (S.A.N., 14-50-00060), the Canadian Foundation for Innovation and Ontario Research Fund (S.-S.B., 11810), the Canadian Stroke Network (S.-S.B.), Heart and Stroke Foundation of Ontario New (S.-S.B., NI6581) and Career (S.-S.B., CI7432) to Investigator Awards, and Start-up Funding from the University of Toronto (S.-S.B.). We thank Dr Paul Dorian for generously donating dog and pig tissue samples. We thank Alexandra Erin Papaelias for graphic design.

## Author contributions

J.T.K., A.S.L., H.Z., R.A.-S. and D.L. collected and analysed the data. J.T.K. prepared the figures. J.T.K., D.L. and S.-S.B. prepared the manuscript. S.O. generated the smooth muscle cell-specific *Cre* mice. S.A.N. generated the floxed TNF mice and provided intellectual expertise. S.P.H., P.H.B. and S.-S.B. supervised the data collection. J.T.K., D.L. and S.-S.B. contributed to the experimental design. All authors reviewed and edited the manuscript.

## Additional information

**Competing interests:** The authors declare no competing financial interests.

