## [Peer Review File · Nature Communications]

Reviewers' comments:

Reviewer #1 (expert in vascular muscle biology)

Remarks to the Author):

This manuscript presents several novel and important results. The data is of uniformly high quality. Many different types of experiments were done in order to test the hypotheses rigorously. The key conclusions are well summarized in Abstract. These include that 1) that membrane bound TNF seems to serve as a constitutive signaling element in myogenic responsiveness, and 2) that interference with mTNF has hemodynamic consequences in living animals that are logically the result of its role in maintaining arterial resistance in certain vascular beds via the myogenic mechanism.

A weakness I see in the report is the assignment of mechanosensor role to mTNF. The ideas and scheme presented seem plausible, but there is no direct evidence that mTNF is a mechanosensor. Alternatively, mTNF could be linked to some other molecular entity that actually serves as the mechanosensor.

A second weakness is the speculations as to the clinical significance of the results. The speculation seems reasonable, but again, the proof in this manuscript is lacking. For example, lines 118 to 128. Specifically, line 122, the 'clinical trials' mentioned are not cited, nor are any details given from those trials about how blood pressure was measured in those trials.

Greater rigor should be used in several such instances throughout the manuscript.

In summary however, this is an impressive and significant work that clearly establishes new key signaling pathways in the control of arterial tone. These pathways must certainly involved in the actions of common biopharmaceuticals administered for treating autoimmune disorders. The results do have important implications for clinical therapy, as stated.

Reviewer #2 (expert in TNFa and cardiovascular biology)

Remarks to the Author:

This is an interesting and well done study that suggests a potentially important role for membrane bound tumor necrosis factor (mTNF) in regulating constitutive myogenic vascular signaling. Further, this study may provide an important insight into one of the mechanisms for the untoward effects of anti-TNF therapy noted in heart failure trials. There are several suggestions for the authors should consider.

1. The authors present indirect data that TNF signaling is important for regulating vascular myogenic tone. In many tissues TNF is not constitutively expressed. Can the authors provide either references or data to show that TNF is constitutively expressed in vascular beds, as well as provide evidence that mTNF is constitutively expressed in vascular beds in the absence of some form of tissue injury.

2. Many of the studies with "reverse signaling" and TNF antagonists have been performed in vitro. The in vivo significance of "reverse signaling" with TNF antagonists is less clear. Thus, the results presented by the authors are potentially very important. The effects of etanercept on reverse signaling in vitro, although described in some (but not all) studies, has been shown to be significantly less than the reverse signaling observed with infliximab or with adalimumab. Given that the clinical results with infliximab in the ATTACH Trial were significantly worse than with etanercept in the RENEWAL Trial, it would be extremely important to confirm the authors' results using infliximab, and repeat the studies that measured diurnal changes in blood pressure. An additional reason why this is important is that in

in vitro etanercept has been shown to neutralize membrane bound lymphotoxin, whereas infliximab and adalimumab do not. Thus, these studies would add specificity to the authors' findings. This additional type of confirmatory study would be helpful in terms of the clinical applicability of the findings.

3. The authors provide data that increasing transmural pressure stimulates p44/42 phosphorylation, elevates intracellular calcium, and stimulates myosin light chain 20 phosphorylation. They then show that these events are sensitive to inhibition with etanercept. It would also be to confirm these studies in the SM-TNF-KO mice, which would exclude any non-specific effects of etanercept (e.g. Fc mediated effects).

4. The diurnal changes in blood pressure are relatively modest, but statistically significant. To complement these studies, the authors should also perform an analysis of heart rate variability in the WT and SM-TNF-KO mice. These data should be readily available to authors. If the change in blood pressure during the day is physiologically significant, the SM-TNK-KO mice should manifest decreased heart rate variability.

5. The drop in blood pressure following intraperitoneal injection of etanercept (lines 114-116; Figure 2B) is dramatic and unexpected. The authors should repeat this experiment using heat inactivated etanercept. Since etanercept is a human chimeric protein, it is possible that the drop in blood pressure is less the effect of etanercept on reverse signaling and myogenic tone than an immune response to the injection of foreign protein. The author's statement on lines 116-117 that etanercept treatment "challenges are already comprised cardiovascular system" cannot be inferred from the injection of intraperitoneal injection of etanercept since etanercept is injected subcutaneously in patients. The authors may consider softening this statement.

6. The authors state on lines 153-155 that TNF is a "mandatory element of myogenic signaling; this non-redundant function is not compensated by other signaling mechanisms when TNF signaling is acutely inhibited." Given as the authors note that the TNF knockout mice undergo a number of myogenic mechanisms to adapt to the inherited loss of TNF activity, and that the time course of this response is not really known, the authors may want to modify this statement by stating that TNF is an important initial element of myogenic signaling during acute TNF inhibition.

Reviewer #3 (expert in vascular muscle biology)

Remarks to the Author:

The present study investigated the role of tumor necrosis factor in myogenic responses in small arteries and in blood pressure control. Mice with smooth muscle TNF gene removal by crossing tamoxifen-treated SMMHC-CreER mice with TNF^{flox/flox} mice and treatment with the TNFR2 mimicking drug etanercept were used. Blood pressure measurements were performed and myogenic tone was measured in isolated cremaster arteries from mice and in gracialis skeletal muscle arteries from a range of species. In addition, molecular studies were performed. The main findings were that deleting the TNF gene in smooth muscle cells or scavenging TNF reduces systemic blood pressure and the myogenic response. The authors propose that reverse signaling of increases in transmural pressure through membrane bound TNF (mTNF) leads to activation of ERK1/2 and SphK1, and that the molecule may function as a mechanosensor. The manuscript is well written with a large number of supportive figures and state of the art methodology. However, there are a series of points which the authors should consider.

1. That there is a role of TNF- α in myogenic tone appears well substantiated by the results presented in the manuscript (Figure 1), while the suggestion that mTNF is actually a mechanosensor seems less apparent. a) Thus, further evidence e.g. PCR showing whether TNFR1 and TNFR2 are actually

expressed in the skeletal smooth muscle would help clarify this issue. b) Moreover, additional studies to clarify a role for TNFR2 and/or TNFR1 in the myogenic response should be performed.

2. Etanercept has lower avidity for tmTNF- α than other TNF- α blockers in clinical use e.g. infliximab, adalimumab, golimumab, and certulizumab. Why was etanercept chosen? Does one of the other blockers have the same effect as etanercept?

3. Lines 152-165, the authors discuss the role of TNF- α in myogenic tone, but fail to consider whether etanercept may affect other well described mechanisms involved in myogenic tone. Again an argument for examining the effect of another blocker.

4. TNF- α is upregulated in heart failure patients have been suggested to be involved in angiotensin II - induced vasoconstriction. However, in the present study all control experiments have mainly been performed with phenylephrine. To clarify whether this pathway which can also lead to activation of ERK1/2, additional studies should be performed in small arteries with angiotensin II in the absence and the presence of etanercept.

5. Different concentrations of etanercept have been applied in the present study, but there are no attempts to establish a dose-relationship to the effects on the myogenic response e.g. low dose etanercept appeared promising in heart failure e.g. Jacobsson-LT et al., J Reumatol 2005, while the clinical studies showing increased mortality were with high doses of etanercept. This is also supported by the present study e.g. in the supplementary Figure 1 despite increased myogenic tone in arteries from heart failure vs. control mice, the differences disappear in the presence of etanercept.

Minor points

1. The choice of the olfactory cerebral arteries is surprising. What is the explanation for performing experiments in these arteries?

2. Fig. 2a is not described in the text.

3. Line 179 "to in"

Responses to Reviewers' Comments:

Reviewer #1

1. **A weakness I see in the report is the assignment of mechanosensor role to mTNF. The ideas and scheme presented seem plausible, but there is no direct evidence that mTNF is a mechanosensor. Alternatively, mTNF could be linked to some other molecular entity that actually serves as the mechanosensor.**

As the reviewer undoubtedly appreciates, the inherent complexity of the experimental system, combined with the small tissue sample size, makes it virtually impossible to collect direct evidence that mTNF is a mechanosensor (i.e., by directly linking physical force to a biologically relevant conformational change in the TNF molecule that alters protein association with TNF or induces a chemical signal).

Nevertheless, our data provide strong evidence that the myogenic signaling cascade is initiated by an mTNF reverse signal and firmly excludes the actions of a soluble ligand. This aspect is significant, because myogenic signaling is perpetually active and thus, proteolytic shedding would rapidly deplete ligand stores. By extension, initiating a reverse signal *without a soluble ligand* can only be achieved if mTNF fulfills the proposed mechanosensor role. In our previous version, we presumed that mTNF tethered to a TNF receptor to form a mechanosensitive pairing.

In this revised version of our manuscript, we have added new data that confirms this presumption by identifying TNF receptors (TNFRs) as the tethering partner for TNF (**Fig. 5m,n**). The complete loss of mTNF-mediated mechanosensitivity in TNFR1/2 knockout animals indicates that mTNF tethering is restricted to TNF receptors. We cannot exclude that other entities are involved in the mechanotransduction process; indeed, accessory proteins would be necessary to transmit the reverse signal¹.

In the revised manuscript, we have expanded our description of the proposed mechanosensitive signaling mechanism (Discussion/Paragraphs 2-3). In the scheme presented (**Fig. 6**), the central mechanosensitive complex transducing the mechanical signal is the mTNF / TNFR pair: both elements are required in order to transduce a physical stimulus.

2. **A second weakness is the speculations as to the clinical significance of the results. The speculation seems reasonable, but again, the proof in this manuscript is lacking. For example, lines 118 to 128. Specifically, line 122, the 'clinical trials' mentioned are not cited, nor are any details given from those trials about how blood pressure was measured in those trials. Greater rigor should be used in several such instances throughout the manuscript.**

We regret the oversight and have duly corrected the citation omissions. With regard to the anti-TNF clinical trials (i.e., ATTACH² and RECOVER, RENAISSANCE, and RENEWAL³), the pertinent publications only provide the initial systemic blood pressure measurements prior to initiating treatment. Thus, the hemodynamic consequences of anti-TNF therapeutics in these trials remain unknown. We have ensured that these clinical trials are properly cited and described appropriately (Discussion/Paragraph 6).

Reviewer #2

1. **The authors present indirect data that TNF signaling is important for regulating vascular myogenic tone. In many tissues TNF is not constitutively expressed. Can the authors provide either references or data to show that TNF is constitutively expressed in vascular beds, as well as provide evidence that mTNF is constitutively expressed in vascular beds in the absence of some form of tissue injury.**

Ideally, western blots would provide direct evidence for constitutive TNF expression in cremaster arteries. Unfortunately, (i) the small amount of tissue, (ii) relatively poor anti-TNF antibodies and (iii) relatively low constitutive TNF expression levels make western blotting impractical, because the signal falls below the necessary detection threshold. The only western blots we have successfully completed with cremaster arteries involve highly abundant proteins: ERK1/2 and MLC₂₀.

As an alternative to western blots, we now provide PCR data that TNF and its receptors TNFR1 and TNFR2 are constitutively expressed at the mRNA level (new data; **Supplementary Fig. 15**). Our acute TNF gene knockout data (Sm-TNF-KO, new data; **Supplementary Fig. 7**) and sTNFR1-Fc fragment data in wild-type and *Tnf*^{-/-} arteries (**Figure 5e-g**) provide functional evidence that mTNF is constitutively expressed under naïve conditions.

2. **Many of the studies with "reverse signaling" and TNF antagonists have been performed in vitro. The in vivo significance of "reverse signaling" with TNF antagonists is less clear. Thus, the results presented by the authors are potentially very important. The effects of etanercept on reverse signaling in vitro, although described in some (but not all) studies, has been shown to be significantly less than the reverse signaling observed with infliximab or with adalimumab. Given that the clinical results with infliximab in the ATTACH Trial were significantly worse than with etanercept in the RENEWAL Trial, it would be extremely important to confirm the authors' results using infliximab, and repeat the studies that measured diurnal changes in blood pressure. An additional reason why this is important is that in vitro etanercept has been shown to neutralize membrane bound lymphotoxin, whereas infliximab and adalimumab do not. Thus, these studies would add specificity to the authors' findings. This additional type of confirmatory study would be helpful in terms of the clinical applicability of the findings.**

We are pleased that the Reviewer asserts that our data describing TNF reverse signaling in resistance arteries are "potentially very important". In order to pursue this mechanistic study in mice, we required an anti-TNF therapeutic that reliably inhibits mouse TNF. From the outset, we strategically selected etanercept (a decoy receptor construct) to pharmacologically block TNF, because it reliably inhibits TNF signalling in mice^{4,5}. Indeed, prior to this investigation, we had successfully utilized etanercept therapeutically in several experimental pathologies⁶⁻¹¹ and a small clinical cohort⁶.

Neutralizing antibodies represent a mechanistically distinct means to inhibit TNF signalling; however, species cross reactivity is problematic. Pharmacological reviews indicate that Infliximab does not cross react with mouse TNF¹²; Adalimumab displays some cross-reactivity, but it has at least 1,000-fold less affinity for murine TNF, relative to human TNF¹³. This aspect seriously handicaps the likelihood that clinically available neutralizing TNF antibodies will possess the necessary efficacy to elicit changes in diurnal blood pressure in mice.

Despite its low affinity for mouse TNF, there are sparse reports that prolonged adalimumab treatment may deliver positive effects in certain murine models of disease (e.g., retinal degeneration and atherosclerosis)^{14,15}. We therefore tested adalimumab, under the cautious premise that it may display efficacy in our experimental systems. Unfortunately, we found no evidence that adalimumab reduces MAP or myogenic responsiveness (**Supplementary Fig. 8**). These results are not entirely surprising, given adalimumab's documented issues in mouse models¹³. These data do, however, demonstrate that human IgG (a component of both etanercept and adalimumab) does not cause an immunological

response that lowers blood pressure; for this reason, we opted to include the negative data in the manuscript.

- 3. The authors provide data that increasing transmural pressure stimulates p44/42 phosphorylation, elevates intracellular calcium, and stimulates myosin light chain 20 phosphorylation. They then show that these events are sensitive to inhibition with etanercept. It would also be to confirm these studies in the SM-TNF-KO mice, which would exclude any non-specific effects of etanercept (e.g. Fc mediated effects).**

Throughout this study, we have consistently observed that etanercept exerts a stronger inhibitory effect than induced gene deletion: these observations apply to the step-wise myogenic tone assessments (**Fig. 3f** vs. **Fig. 2m**), single-step myogenic responses (**Fig. 3p** vs. **Supplementary Fig. 7b**) and the newly added molecular data (**Fig. 3m** vs. **Supplementary Fig. 7c**). There are two potential factors that contribute to this differential effect: first, as the reviewer is undoubtedly aware, inducible gene deletion models do not always achieve complete gene deletion (e.g., cell-to-cell differences in gene deletion)¹⁶. The complexity of our experimental system, coupled with the fact that TNF is not a highly expressed protein, makes it virtually impossible to accurately determine the extent and homogeneity of TNF gene deletion throughout the vessel wall. Second, the chronic and relatively slow removal of TNF expression could potentially engage adaptations: these compensatory responses could potentially amplify residual TNF signaling (i.e., if gene deletion is incomplete) or the recruitment of alternative pressure-sensitive pathways (i.e., as is observed in germline knockout mice). If adaptations are occurring, they are clearly incomplete within the timeframe of our assessments: thus, only a small fraction of the original response could be explained by such an effect.

In summary, inducible gene deletion is a highly-specific approach to define critical signaling elements within a pathway; however, direct comparisons with respect to magnitude of inhibition should be treated with due caution. In this context, we agree that substantiating our key molecular observations in Sm-TNF-KO would strengthen our conclusions: we therefore chose to examine ERK1/2 phosphorylation in arteries isolated from Sm-TNF-KO mice, since we have characterized this response as a proximal element in the myogenic signaling cascade (i.e., a response that is proximal to TNF)¹⁷. As expected, smooth muscle cell TNF gene deletion attenuated pressure-stimulated ERK1/2 phosphorylation (**Supplementary Fig. 7**), confirming that this upstream response is specifically activated through a TNF-derived signal.

- 4. The diurnal changes in blood pressure are relatively modest, but statistically significant. To complement these studies, the authors should also perform an analysis of heart rate variability in the WT and SM-TNF-KO mice. These data should be readily available to authors. If the change in blood pressure during the day is physiologically significant, the SM-TNK-KO mice should manifest decreased heart rate variability.**

We agree that analyzing heart rate variability (HRV) would complement our diurnal mean arterial pressure recordings. Unfortunately, our radio-telemetric recordings are binned into 30 second blocks. Since beat-to-beat recordings are required to calculate HRV, our data do not permit HRV calculations.

5. **The drop in blood pressure following intraperitoneal injection of etanercept (lines 114-116; Figure 2B) is dramatic and unexpected. The authors should repeat this experiment using heat inactivated etanercept. Since etanercept is a human chimeric protein, it is possible that the drop in blood pressure is less the effect of etanercept on reverse signaling and myogenic tone than an immune response to the injection of foreign protein. The author's statement on lines 116-117 that etanercept treatment "challenges are already comprised cardiovascular system" cannot be inferred from the injection of intraperitoneal injection of etanercept since etanercept is injected subcutaneously in patients. The authors may consider softening this statement.**

As requested, we repeated our etanercept injection experiments using heat-inactivated etanercept. As expected, no reduction in blood pressure was observed (**Supplementary Fig. 4**). We therefore conclude that the drop in MAP following i.p. ETN injection is not an immune response to a foreign protein. As suggested by the reviewer, we have revised our statement regarding etanercept “challenging an already compromised cardiovascular system” (Discussion/Paragraph 6).

6. **The authors state on lines 153-155 that TNF is a "mandatory element of myogenic signaling; this non-redundant function is not compensated by other signaling mechanisms when TNF signaling is acutely inhibited." Given as the authors note that the TNF knockout mice undergo a number of myogenic mechanisms to adapt to the inherited loss of TNF activity, and that the time course of this response is not really known, the authors may want to modify this statement by stating that TNF is an important initial element of myogenic signaling during acute TNF inhibition.**

As suggested, we have modified the statement highlighted by the reviewer (Discussion/Paragraph 1).

Reviewer #3

1. **That there is a role of TNF- α in myogenic tone appears well substantiated by the results presented in the manuscript (Figure 1), while the suggestion that mTNF is actually a mechanosensor seems less apparent. a) Thus, further evidence e.g. PCR showing whether TNFR1 and TNFR2 are actually expressed in the skeletal smooth muscle would help clarify this issue. b) Moreover, additional studies to clarify a role for TNFR2 and/or TNFR1 in the myogenic response should be performed.**

We are pleased that the Reviewer is convinced that TNF plays key role in myogenic signaling. As the reviewer undoubtedly appreciates, the inherent complexity of the experimental system, combined with the small tissue sample size, makes it virtually impossible to collect direct evidence that mTNF is a mechanosensor (i.e., by directly linking physical force to a biologically relevant conformational change in the TNF molecule that alters protein association with TNF or induces a chemical signal).

As requested, we have added (a) PCR data demonstrating the expression of TNFR1 and TNFR2 in cremaster arteries (**Supplementary Fig. 15**) and (b) functional measurements in cremaster arteries isolated from TNFR1/TNFR2 double-knockout mice (**Fig. 5m,n**). The latter data identify TNF receptors (TNFRs) as the tethering partner for TNF. The complete loss of mTNF-mediated mechanosensitivity in TNFR1/2 knockout animals indicates that mTNF tethering is restricted to TNF receptors.

In the revised manuscript, we have expanded our description of the proposed mechanosensitive signaling mechanism (Discussion/ Paragraphs 2-3). In the scheme presented (**Fig. 6**), the central mechanosensitive complex transducing the mechanical signal is the mTNF / TNFR pair: both elements are required in order to transduce a physical stimulus.

2. **Etanercept has lower avidity for tmTNF- α than other TNF- α blockers in clinical use e.g. infliximab, adalimumab, golimumab, and certulizumab. Why was etanercept chosen? Does one of the other blockers have the same effect as etanercept?**

As discussed in our response to Reviewer #2 (comment 2), in order to pursue this mechanistic study in mice, we required an anti-TNF therapeutic that reliably inhibits mouse TNF. From the outset, we strategically selected etanercept (a decoy receptor construct) to pharmacologically inhibit TNF, because it reliably inhibits TNF signalling in mice^{4,5}. Indeed, prior to this investigation, we successfully utilized etanercept therapeutically in several experimental pathologies⁶⁻¹¹ and a small clinical cohort⁶.

Neutralizing antibodies represent a mechanistically distinct means to inhibit TNF signalling; however, species cross reactivity is problematic. Pharmacological reviews indicate that Infliximab does not cross react with mouse TNF¹²; Adalimumab displays some cross-reactivity, but it has at least 1,000-fold less affinity for murine TNF, relative to human TNF¹³. This aspect seriously handicaps the likelihood that clinically available neutralizing TNF antibodies will possess the necessary efficacy to elicit changes in diurnal blood pressure in mice.

Despite its low affinity for mouse TNF, there are sparse reports that prolonged adalimumab treatment may deliver positive effects in certain murine models of disease (e.g., retinal degeneration and atherosclerosis)^{14,15}. We therefore tested adalimumab, under the cautious premise that it may display efficacy in our experimental systems. Unfortunately, we found no evidence that adalimumab reduces MAP or myogenic responsiveness (**Supplementary Fig. 8**). These results are not entirely surprising, given adalimumab's documented issues in mouse models¹³.

In the revised manuscript, we elaborate on this issue (Discussion/Paragraph 7) and we include the negative result with adalimumab in the manuscript.

3. Lines 152-165, the authors discuss the role of TNF- α in myogenic tone, but fail to consider whether etanercept may affect other well described mechanisms involved in myogenic tone. Again an argument for examining the effect of another blocker.

Since neutralizing anti-TNF antibodies display limited efficacy against mouse TNF, additional “blocker” options are limited. Our control experiments (some control are newly-added to this revised version) indicate that etanercept’s actions are specific: (i) denatured/boiled etanercept does not attenuate myogenic responsiveness *in vitro* (**Fig. 3I**) nor does it reduce MAP *in vivo* (**Supplementary Fig. 4**); and (ii) lacks efficacy in TNF knockout mice (**Fig. 3i,j**). Taken together, there are minimal concerns that off-target effects confound our study’s interpretations. Consequently, we assert that our characterization that TNF is an important regulator of myogenic tone in physiological, non-pathological setting is appropriate.

4. TNF- α is upregulated in heart failure patients have been suggested to be involved in angiotensin II-induced vasoconstriction. However, in the present study all control experiments have mainly been performed with phenylephrine. To clarify whether this pathway which can also lead to activation of ERK1/2, additional studies should be performed in small arteries with angiotensin II in the absence and the presence of etanercept.

We recognize that angiotensin II may modulate cremaster artery myogenic tone *in vivo*. The overwhelming majority of experimental data we present was collected in an *in vitro* experimental system of isolated arteries that excludes systemic influences on vasoconstrictive mechanisms. In this context, the TNF-dependent mechanism documented in this investigation is independent of angiotensin II. This conclusion does not preclude a relationship between angiotensin II and TNF *in vivo*; however, this aspect is beyond the scope of the present investigation. We selected phenylephrine for our viability and general contractility control, because it is a reliable, potent and well characterized vasoconstrictor; this has been our standard contractility control for all of our previous investigations.

5. Different concentrations of etanercept have been applied in the present study, but there are no attempts to establish a dose-relationship to the effects on the myogenic response e.g. low dose etanercept appeared promising in heart failure e.g. Jacobsson-LT et al., J Reumatol 2005, while the clinical studies showing increased mortality were with high doses of etanercept. This is also supported by the present study e.g. in the supplementary Figure 1 despite increased myogenic tone in arteries from heart failure vs. control mice, the differences disappear in the presence of etanercept.

We selected the etanercept concentration for our *in vitro* mouse model based on an initial dose-response relationship that we constructed (**Fig. 3g**); subsequent experiments utilized 300 μ g/ml throughout. In mouse cremaster arteries, this concentration effectively abolishes myogenic reactivity, while preserving vasoconstriction to phenylephrine. The only instance where we used a different etanercept concentration *in vitro* (10 μ g/ml) was for hamster arteries, where our previous work dictated the effective dose⁷.

Establishing a dose-response relationship *in vivo* will not add to our mechanistic interpretations: in the context of a basic science investigation, our animal ethics committee would not approve such a profound use of animals for a dosing assessment, without a strong rationale for its necessity in the mechanistic interpretation. We accept that dosing information may be required for clinical application; however, this is clearly beyond the scope of the present investigation.

Minor points

1. The choice of the olfactory cerebral arteries is surprising. What is the explanation for performing experiments in these arteries?

Olfactory cerebral arteries were selected because: (i) unlike other cerebral arteries we have investigated (e.g., posterior cerebral arteries)⁷, olfactory arteries display significant myogenic tone under non-pathological settings; and (ii) forward signaling (sTNF/TNFR) dominates TNF's modulation of myogenic signaling¹⁰. These attributes permitted control experiments relevant to mTNF-mediated reverse signaling. Our data suggest, at least for the vascular beds assessed, that mTNF reverse signaling and sTNFR/TNFR forward signaling are mutually exclusive mechanisms that do not simultaneously modulate myogenic tone. We have adjusted the manuscript to better describe that rationale for olfactory artery selection (Results/Myogenic signaling routes through membrane-bound TNF/Paragraph 2).

2. Fig. 2a is not described in the text.

We apologize for the oversight. The manuscript has been adjusted accordingly (Results/Etanercept reduces blood pressure and myogenic signaling/Paragraph 1).

3. Line 179 "to in"

We have corrected this typographical error.

REFERENCES

1. Watts, A. D. *et al.* A casein kinase I motif present in the cytoplasmic domain of members of the tumour necrosis factor ligand family is implicated in 'reverse signalling'. *EMBO J* **18**, 2119-2126 (1999).
2. Chung, E. S. *et al.* Randomized, double-blind, placebo-controlled, pilot trial of infliximab, a chimeric monoclonal antibody to tumor necrosis factor- α , in patients with moderate-to-severe heart failure: results of the anti-TNF Therapy Against Congestive Heart Failure (ATTACH) trial. *Circulation* **107**, 3133-3140 (2003).
3. Mann, D. L. *et al.* Targeted anticytokine therapy in patients with chronic heart failure: results of the Randomized Etanercept Worldwide Evaluation (RENEWAL). *Circulation* **109**, 1594-1602 (2004).
4. Kruglov, A. A. *et al.* Modalities of experimental TNF blockade in vivo: mouse models. *Adv Exp Med Biol* **691**, 421-431 (2011).
5. Drutskaya, M. S. *et al.* Experimental models of arthritis in which pathogenesis is dependent on TNF expression. *Biochemistry (Mosc)* **79**, 1349-1357 (2014).
6. Scherer, E. Q. *et al.* Tumor necrosis factor- α enhances microvascular tone and reduces blood flow in the cochlea via enhanced sphingosine-1-phosphate signaling. *Stroke* **41**, 2618-2624 (2010).
7. Yang, J. *et al.* Proximal cerebral arteries develop myogenic responsiveness in heart failure via tumor necrosis factor- α -dependent activation of sphingosine-1-phosphate signaling. *Circulation* **126**, 196-206 (2012).
8. Meissner, A. *et al.* Tumor necrosis factor- α -mediated downregulation of the cystic fibrosis transmembrane conductance regulator drives pathological sphingosine-1-phosphate signaling in a mouse model of heart failure. *Circulation* **125**, 2739-2750 (2012).
9. Meissner, A. *et al.* Tumor Necrosis Factor- α Underlies Loss of Cortical Dendritic Spine Density in a Mouse Model of Congestive Heart Failure. *J Am Heart Assoc* **4**, (2015).
10. Yagi, K. *et al.* Therapeutically Targeting Tumor Necrosis Factor- α /Sphingosine-1-Phosphate Signaling Corrects Myogenic Reactivity in Subarachnoid Hemorrhage. *Stroke* (2015).
11. Sauvé, M. *et al.* Tumor Necrosis Factor/Sphingosine-1-Phosphate Signaling Augments Resistance Artery Myogenic Tone in Diabetes. *Diabetes* **65**, 1916-1928 (2016).
12. Martin, P. L. & Bugelski, P. J. Concordance of preclinical and clinical pharmacology and toxicology of monoclonal antibodies and fusion proteins: soluble targets. *Br J Pharmacol* **166**, 806-822 (2012).
13. FDA Adalimumab (Humira). <http://www.fda.gov/downloads/Drugs/DevelopmentApprovalProcess/HowDrugsareDevelopedandApproved/ApprovalApplications/TherapeuticBiologicApplications/ucm092772.pdf> (2002).
14. de la Cámara, C. M. -F. *et al.* Adalimumab reduces photoreceptor cell death in a mouse model of retinal degeneration. *Scientific reports* **5**, (2015).
15. Zhu, L. *et al.* Loss of Macrophage Low-Density Lipoprotein Receptor-Related Protein 1 Confers Resistance to the Antiatherogenic Effects of Tumor Necrosis Factor- α Inhibition. *Arterioscler Thromb Vasc Biol* **36**, 1483-1495 (2016).
16. Sauer, B. Inducible gene targeting in mice using the Cre/lox system. *Methods* **14**, 381-392 (1998).

17. Lidington, D. *et al.* The phosphorylation motif at serine 225 governs the localization and function of sphingosine kinase 1 in resistance arteries. *Arterioscler Thromb Vasc Biol* **29**, 1916-1922 (2009).

Reviewers' comments:

Reviewer #1 (Remarks to the Author):

My original comments were concerned primarily with two issues; the problem of direct evidence of a mechanosensor role for mTNF, and a lack of appropriate citations in support of certain statements.

The addition of the TNFR experiment is significant, certainly represents a major effort to address the issue, and, in my opinion was largely successful. Fig. 6 has been modified appropriately. The manuscript is strengthened considerably, and the weakness I identified has been obviated.

Citations are now appropriate.

Reviewer #2 (Remarks to the Author):

None

Reviewer #3 (Remarks to the Author):

The present study investigated the role of tumor necrosis factor in myogenic responses in small arteries and in blood pressure control. The authors propose that reverse signaling of increases in transmural pressure through membrane bound TNF (mTNF) leads to activation of ERK1/2 and SphK1, and that the molecule may function as a mechanosensor. The authors have performed additional experiments and clarified some of the points raised.

The authors ignore the role of angiotensin II and AT1 receptors in the myogenic response, although several research groups have found a role for AT1 receptors in the myogenic response (Hong et al., J Physiol, 2016; 594.23, 594-623; Schleifenbaum et al., Circ Res, 2014; 115, 263-272). Moreover, there is evidence that angiotensin II can be formed in the vascular wall of small arteries.

Angiotensin II/AT1 receptor activation also leads to increased ERK1/2 phosphorylation and as well as stimulation of TNF α etc. This raises the question where in the chain of signaling events involved in the myogenic response membrane-bound TNF is. As mentioned in connection with the first evaluation of the manuscript a simple series of in vitro experiments performing angiotensin response curves in the absence and the presence of etanercept would help in clarifying that.

Responses to Reviewers' Comments:

Reviewer #3

1. **Angiotensin II/AT1 receptor activation also leads to increased ERK1/2 phosphorylation and as well as stimulation of TNF α etc. This raises the question where in the chain of signaling events involved in the myogenic response membrane-bound TNF is. As mentioned in connection with the first evaluation of the manuscript a simple series of *in vitro* experiments performing angiotensin response curves in the absence and the presence of etanercept would help in clarifying that.**

As requested, we performed the suggested angiotensin II (Ang II) dose-response experiment. We observed that Ang II induces relatively modest vasoconstriction, even at a supra-physiological concentration (1 $\mu\text{mol/L}$). Etanercept reduces basal tone, consistent with our experiments (i.e., phenylephrine dose-response curve; Figure 3H), but does not affect Ang II-stimulated vasoconstriction. In fact, when basal tone is accounted for, Ang II responses are marginally augmented.

Although angiotensin type I receptors have been implicated as myogenic modulators¹⁻³, etanercept's lack of effect on Ang II-stimulated vasoconstriction implies that angiotensin type I receptor signaling is not linked to the TNF-dependent mechanism we describe in the present manuscript. Further, the relatively modest vasoconstriction induced by Ang II suggests that it is not a prominent regulator of vascular tone in the arteries studied here. The role for angiotensin and angiotensin receptors differs across species and vascular beds²: these differences may explain why its role appears to be absent in our model.

In the revised manuscript, we have added the new Ang II dose-response data to *Supplementary Figure 6*. We have briefly described the results and the implication in the Results subsection.

References:

1. Mederos y Schnitzler, M. *et al.* Gq-coupled receptors as mechanosensors mediating myogenic vasoconstriction. *EMBO J* **27**, 3092-3103 (2008).
2. Hong, K. *et al.* Mechanical activation of angiotensin II type 1 receptors causes actin remodelling and myogenic responsiveness in skeletal muscle arterioles. *J Physiol* **594**, 7027-7047 (2016).
3. Schleifenbaum, J. *et al.* Stretch-activation of angiotensin II type 1a receptors contributes to the myogenic response of mouse mesenteric and renal arteries. *Circ Res* **115**, 263-272 (2014).

REVIEWERS' COMMENTS:

Reviewer #3 (Remarks to the Author):

The present manuscript the authors have included experiments showing the effect of angiotensin II in the absence and the presence of etanercept and show a minor role for TNFalpha in angiotensin II - induced vasoconstriction. The manuscript has been revised accordingly.